# Off-fault damage controls near-surface rupture behaviour in soft sediments

Nicola De Paola [1] ✉, Rachael J. Bullock[1], Robert E. Holdsworth [1], Shmuel Marco[2] & Stefan Nielsen [1]

Surface-rupturing faults represent some of the most devastating examples of earthquake-related hazards. Near surface rupture behaviour is poorly understood, due to limited information concerning the amount of off-fault energy dissipated in a volume surrounding the propagating rupture tip. Here, we show that the energy dissipated by off-fault damage in shallow seismic faults can slow down and arrest ruptures in the near surface regions. We integrate experimental results with field measurements from near-surface seismic faults in the Dead Sea, to estimate that off-fault damage is up to 85% of the total fracture energy dissipated during a $M_w \approx 6$ rupture events. Analytical modeling of the studied faults demonstrates that increased fracture energy from off-fault damage energy dissipation significantly slows or halts rupture propagation near the surface. Here, we plausibly explain the slow rupture velocities and low radiation efficiencies observed in shallow ruptures in soft sediments.

Seismological observations from some large/moderate magnitude earthquakes show that ruptures can nucleate[1] and propagate with a large amount of slip in the shallow regions of fault zones[2–5], where clay-rich soft sediments are often present[6]. Surface-rupturing faults represent some of the most devastating examples of earthquake-related hazards[7]. Nevertheless, their rupture behaviour is still poorly understood, due to uncertainties concerning the amount of energy dissipated during rupture propagation[8,9].

The potential energy released during an earthquake is partitioned into the fracture energy dissipated by on-fault and off-fault processes during rupture propagation, the energy dissipated as frictional heat and the radiated energy[8–10]. Geological[11], experimental[12,13] and numerical modelling[14–16] estimates suggest that the energy dissipated by inelastic, off-fault deformation represents a significant portion of the total energy dissipated in a volume around the rupture tip[9]. Quantifying on- and off-fault energy dissipation during rupture propagation is of paramount importance, as it controls the rupture velocity and peak slip velocity of an earthquake, as well as the amount of radiated energy that produces hazardous ground acceleration[8,9].

Great uncertainties exist about the structure and dominant deformation mechanisms of shallow, active faults in soft sediment[17–20]. This is due to the rare preservation of earthquake surface ruptures,

which are rapidly degraded by surface weathering processes, or are inaccessible, such as below sea level in subduction zones[6]. Hence, constraining the absolute and relative contribution of on-fault and off-fault energy dissipation to the energy balance of near-surface ruptures in soft sediments is particularly challenging[9], leaving many fundamental questions still unanswered. For example, does the structure of a fault influence how energy is lost during an earthquake, by controlling whether it is dissipated along the slipping fault itself or in the surrounding damaged rocks (e.g., energy sinks related to on-fault vs. off-fault deformation)? Do shallow faults in soft sediment act as a brake during rupture propagation or do they lubricate and weaken to favour large slip patches at shallow depths?

Here, we present field data that characterise the fault zone architecture and associated on- and off-fault deformation of near-surface, co-seismic faults in the Dead Sea region. These field measurements, integrated with experimental results, are then used to constrain the fracture energy dissipated both on- and off-fault during a single rupture event. Based on these results, we propose a simple model demonstrating that off-fault energy dissipation is significant and can help explain key geophysical characteristics of shallow ruptures in soft sediments, such as slow rupture velocity and low radiation efficiency[21,22].

[1]Department of Earth Sciences, Durham University, Durham, UK. [2]Department of Geophysics, Tel-Aviv University, Tel Aviv, Israel. ✉e-mail: nicola.de-paola@durham.ac.uk

## Results

### Syn-sedimentary faulting in the Masada Fault Zone (MFZ)

The Masada Fault Zone (MFZ) forms part of a system of normal faults that bound the western side of the Dead Sea Basin, a large pull-apart graben located between two segments of the Dead Sea transform fault, the boundary between the Arabia and Sinai plates[23] (Fig. 1a). The Dead Sea fault has an average long-term strike-slip motion of ~5 mm/year, whilst normal faulting is estimated to account for ~10% of the total fault displacement[24]. The 1927 $M_L = 6.2$ Jericho earthquake was the most recent, large event located close to the Dead Sea[25] (Fig. 1a).

The MFZ has an overall N-S strike, sub-parallel to the main graben-bounding structures (Fig. 1a–c). The faults cut through up to 40-m-thick Pleistocene Lisan Formation sediments[26], which were deposited between $63 \pm 7$ Ka and 18 Ka[27] in Lake Lisan, the precursor to the Dead Sea. The Lisan Formation sediments were subaqueous at the time of deformation and are very poorly lithified, having never been buried deeper than a few tens of metres[26]. In the study area, the upper part of the Lisan Formation sequence is unfaulted (Fig. 1b), indicating that the MFZ has been inactive since ~20-25 Ka[18]. This is due to the migration and progressive localisation of activity towards the centre of the Dead Sea Basin[28]. Ground-penetrating radar and seismic reflection surveys have traced the MFZ faults down to depths of at least 250–300 m[29], well below the base of the Lisan Formation. This shows that they are of tectonic origin, rather than due to syn-depositional gravitational slumping processes[30] (Fig. 1c).

### Field evidence for syn-sedimentary co-seismic slip

Canyon incision during periods of the Dead Sea level retreat has led to the formation of excellent and highly accessible cliffs up to 40 m high, where the MFZ is exposed (Fig. 2a). The Lisan Formation is a varved sequence comprising alternating laminae (<3 mm thick) of white aragonite, precipitated during summer months, and dark clastic detritus deposited during winter flash flooding events (Fig. 2b). The clastic laminae comprise sub-angular to sub-rounded grains of calcite, dolomite, minor quartz and feldspars, with maximum grain dimensions of up to ~100 µm, plus up to ~70% clay—a mixture of kaolinite, montmorillonite and illite (Fig. 2c; Supplementary Notes in Supplementary Information). The aragonite crystals are acicular (widths up to 2 µm and lengths up to 15 µm) and are commonly arranged in radial, rosette-like structures with diameters of up to 30 µm (Fig. 2d).

The thickening of sedimentary layers observed in the hanging walls of many exposed faults, compared to the equivalent layers in the footwalls, shows that they are syn-sedimentary (Fig. 3a). Ancient surface ruptures along the MFZ are associated with 'seismite' layers (Fig. 3a, b) created by co-seismic shaking-induced gravitational instabilities (e.g., breccias/mixed layers and slumped layers)[17,18,26,31]. Breccia layers contain a chaotic mixture of brecciated, mm- to cm-scale host-rock laminae set in a matrix of ultrafine-grained sand and mud (Fig. 3a, b). They are up to 1 m thick and are laterally extensive over distances of hundreds of metres[17]. The formation of upward-fining breccia layers has been attributed to the shaking of sediments at the lakebed during an earthquake, which are thrown into suspension and subsequently re-deposited[17]. Field observations and paleoseismological studies[18] have revealed distinct seismic clusters along the MFZ, each characterised by a few tens up to approximately 60 cm of co-seismic slip—the most recent occurring around 25 kyr ago. These events are inferred to have been at least magnitude $M = 6$, based on empirical relationships between fault slip and earthquake magnitude[32]. Additional support comes from seismological studies[33,34], which show that surface ruptures, like those observed along the MFZ, are typically associated with earthquakes of magnitude ≥5.5–6. Furthermore, theoretical modelling[35] suggests that sediment liquefaction in the slumped (Fig. 3a) and breccia layers (Fig. 3a, b) requires earthquake magnitudes ≥6, reinforcing the conclusion that events of $M \geq 6$ had sufficient energy to generate the seismites and related structures

identified in the MFZ. These findings establish a lower bound on event magnitudes for the MFZ. Consequently, the syn-sedimentary faults and associated seismites observed in both outcrops and the deep Dead Sea drill core[36] constitute an almost continuous, 220,000-year palaeo-seismic record for the Dead Sea Graben.

### On-fault deformation of syn-sedimentary co-seismic faults

We studied syn-sedimentary faults of the MFZ, with displacement ranging between 100 and 330 cm. The faults cutting the Lisan Formation typically preserve a fault core–damage zone structure. Here, we consider on-fault deformation as the highly localised co-seismic slip accommodated in the fault core within continuous principal slip zones (PSZs) and the interconnected, densely clustered deformation bands (e.g., subsidiary slip surfaces (SSSs) in Fig. 3a, c).

In the studied faults, the fault core width ranges from about 1 mm to 30 cm thick, where branches are present. The fault core structure is complex and made of PSZs and SSSs that branch from the PSZ (e.g., fault LIF2.2 in Fig. 3a; Supplementary Figs. 4 and 5). In the fault cores, fault offsets are mainly localised within a sharp, up to a few centimetres thick PSZ, with a relatively straight principal slip surface bounding it on one side (Fig. 3a, c). At the hand specimen scale, PSZs comprise ultrafine-grained grey to pale brown coloured silty gouges bounded by sharp slip surfaces (Fig. 4a). Scanning electron microscopy (SEM) images of thin sections show that shear localisation occurs in the PSZ adjacent to the PSS, as shown by the preferred alignment of acicular aragonite clasts (Fig. 4b) and by the presence of 'rolled' clasts with a cortex of clay (Fig. 4c, d). Clasts in the PSZ are not cut by fault discontinuities and there is little obvious fracturing or reduction in grain size of the material (Fig. 4b). It is particularly striking that even within the PSZ, some of the aragonite crystals retain their delicate radial, rosette-like arrangements of needles during deformation (Fig. 4c).

Overall, microstructural observations suggest that very little grain abrasion or brittle fracturing has occurred in the PSZ, and that most grains have simply slid or rolled past one another during fault-related deformation. Thus, during co-seismic slip, the dominant deformation mechanism along the MFZ was grain-scale particulate flow[37].

### Off-fault deformation of syn-sedimentary co-seismic faults

We attribute off-fault deformation to the distributed deformation accommodated by isolated shear structures in the damage zones, which are not physically connected to those in the fault core (Figs. 5 and 6).

Off-fault deformation within damage zones adjacent to each fault core is mainly accommodated by brittle structures, such as deformation bands, hybrid extensional-shear fractures and tensile fractures (Fig. 5a, b). Deformation bands display similar orientations to the master faults (Fig. 6a(i, ii)), ranging in length from a few mm to tens of cm. They show offsets of up to a few centimetres, which are accommodated by zones of shear, typically <<1 mm wide (DB in Fig. 6b, c). Hybrid extensional-shear fractures are typically several tens of cm long with up to a few mm of displacement. They are characterised by either en-echelon or stepped geometries, with several dilational jogs and restraining bends along their lengths (Fig. 6d). They display a greater scatter of orientations compared to the deformation bands (Fig. 6a(iii)). At this and many other locations in the Masada area, pervasive sub-vertical tensile fractures intersect the upper part of the Lisan Formation, whereas the faults do not. They have a wide range of orientations (Fig. 6a(iv)) and do not appear to spatially relate in a systematic way to the location of major faults (Fig. 5b). Hence, tensile fractures appear to post-date faulting due to the progressive migration of extension towards the centre of the Dead Sea basin[38]. These structures have been instead related to sediment unloading during periods of water-level retreat, or to local salt diapir intrusion[38].

Measurements taken along a transect oriented perpendicular to fault strike (Fig. 5a) show a decrease in the frequency of deformation

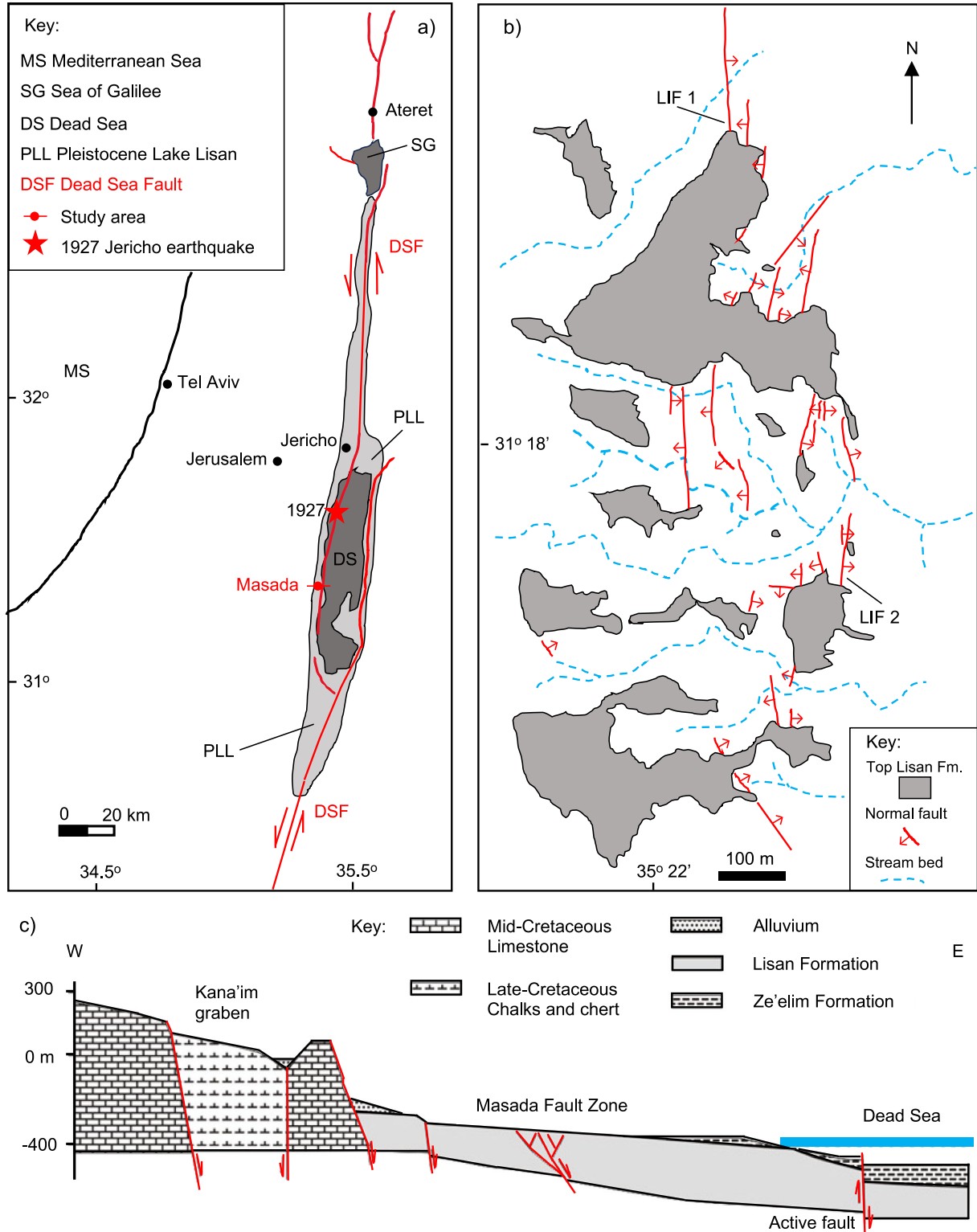

**Fig. 1 | The Masada Fault Zone. a** Location map, showing the main segments of the Dead Sea Fault (DSF) and the maximum extent of Lake Lisan at 26 Ka (shaded pale grey). Adapted from Marco and Agnon[18], with permission from Elsevier. The inferred epicentre of the 1927 Jericho earthquake[25] is indicated by the red star. **b** Map of the Masada Fault Zone (MFZ). LIF 1 and LIF 2 refer to the location of the studied Lisan Fault 1 and 2, respectively. Adapted from Marco and Agnon[18], with permission from Elsevier. **c** Schematic cross-section across the Masada locality highlighted in a), showing the context of the MFZ in relation to the main graben-bounding normal faults and the currently active strand of the Dead Sea fault. Adapted from Marco[28], with permission from Elsevier.

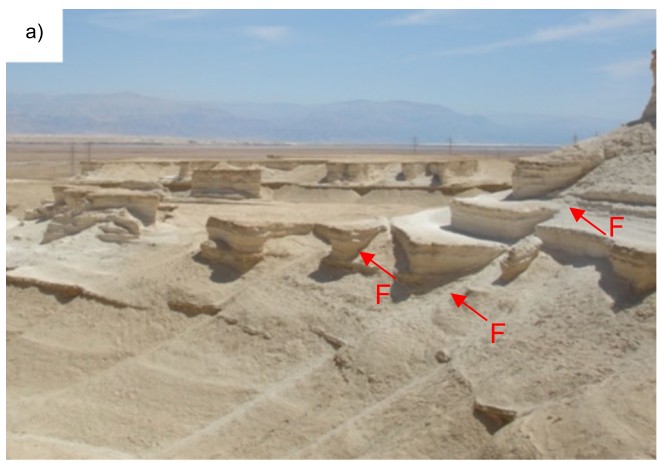

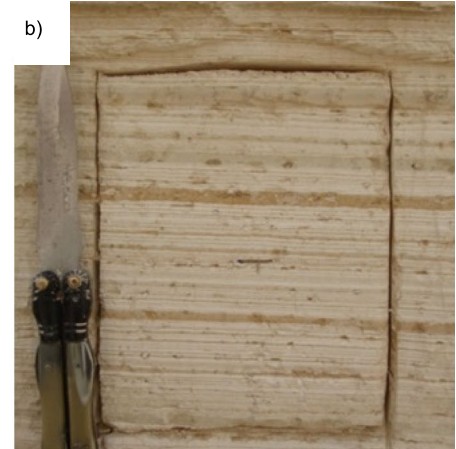

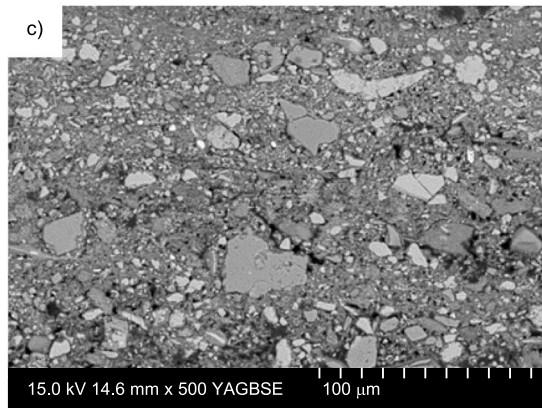

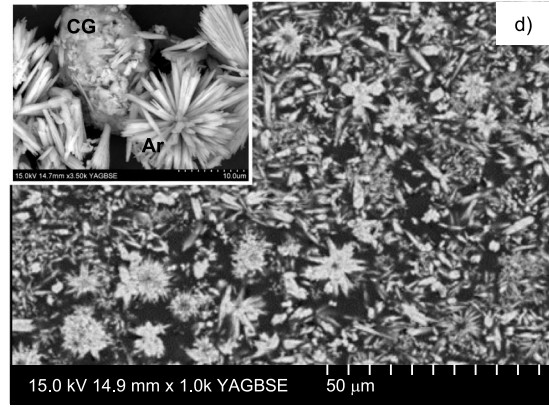

**Fig. 2 | The Lisan Formation sediments. a** Poorly lithified cliffs of the Lisan Formation exposed on the shores of the Dead Sea, with some fault traces highlighted (F). Photo is taken looking east towards the Dead Sea, with Jordan in the far distance. **b** Undeformed Lisan sediment as observed in the field, showing alternating aragonite (white laminae) and clastic (darker grey and brown laminae) layers. **c** SEM image of an undeformed clastic layer, comprising a chaotic mixture of clasts— predominantly quartz, calcite, aragonite and dolomite—in a clay-rich matrix. **d** SEM images of an undeformed aragonite layer. Inset shows a higher magnification SEM image of an aragonite layer showing clay aggregate clasts (CG) and acicular aragonite crystals with sharp, pointed terminations, sometimes arranged in radiating aggregates resembling rosettes (Ar).

bands and hybrid extensional-shear fractures away from the main faults, and an increase in damage zone width with fault displacement (Fig. 5b). Conversely, post-faulting tensile fractures do not appear to spatially relate in a systematic way to the location of major faults (Fig. 5b). Fracture density is also high where the damage zones of two adjacent faults overlap (e.g., faults LIF1.1 and LIF1.3 in Fig. 5b). Overall, these observations are consistent with theoretical predictions of wider damage zones being generated by more energetic, larger slip events[39–41].

## Co-seismic fault strength

To constrain co-seismic fault strength and deformation mechanisms of the MFZ, we performed a series of shear experiments at seismic slip rates. Experiments were performed on room-humidity and Dead Sea brine-saturated Lisan Formation gouges (34.2% salinity), containing between 40 and 60% aragonite (grain size <15 μm), 30–50% clay (mixed layer illite-smectite and kaolinite), and up to 10% quartz, gypsum and calcite (grain size <100 μm) (Supplementary Notes). Microstructures recovered from experimentally deformed samples have also been compared with those observed from natural, seismic fault zones.

We performed 14 friction experiments at room temperature, seismic (1.3 m s$^{-1}$) slip rates and normal loads ranging from 1 to 18 MPa (Supplementary Methods; Supplementary Table 1). The 1 MPa normal load experiments correspond to ~50 m burial depth of sediment with no overlying water column, or surficial sediments with an overlying water column of ~100 m depth. They are therefore representative of the ambient conditions during deformation of the naturally observed Lisan Formation faults. The 9–18 MPa experiments correspond to a sediment burial depth between ~0.5 and 1 km. These experiments allow us to assess how sensitive co-seismic fault strength is to burial depth in the uppermost part of the crust.

All samples sheared at 1.3 m s$^{-1}$ show a dynamic weakening, with friction μ decreasing from peak values $\mu_p$ to low steady-state values $\mu_{ss}$ over a slip-weakening distance $D_w$[42] (Fig. 7a; Supplementary Fig. 9 and Supplementary Table 1). The brine-saturated gouges produce initial peak values of frictional strength in the range $\mu_p = 0.30$–0.36 at the onset of slip (Fig. 7a). They then undergo an immediate slip weakening, over displacements of <6 cm, to very low values of steady-state friction, $\mu_{ss} = 0.10$–0.11 (Fig. 7a). In contrast to the brine-saturated gouges, the room-humidity gouges undergo an initial phase of slip-hardening to attain a peak friction value $\mu_p = 0.79$, before weakening to steady-state values of 0.22 over a slip-weakening distance $D_w$ of 1.67 m (Fig. 7a). These results demonstrate that brine-saturated gouges weaken more significantly and more rapidly than those at room humidity, exhibiting the characteristic instantaneous weakening at slip onset[43–45] (Fig. 7a).

The microstructures of the sheared brine-saturated gouges show the development of incipient slip surfaces close to the top of the sample (Fig. 7b). Similar to what has been observed in the natural gouges of the MFZ, the brine-saturated gouges deformed during the high-velocity experiments show minimal evidence of fracturing and

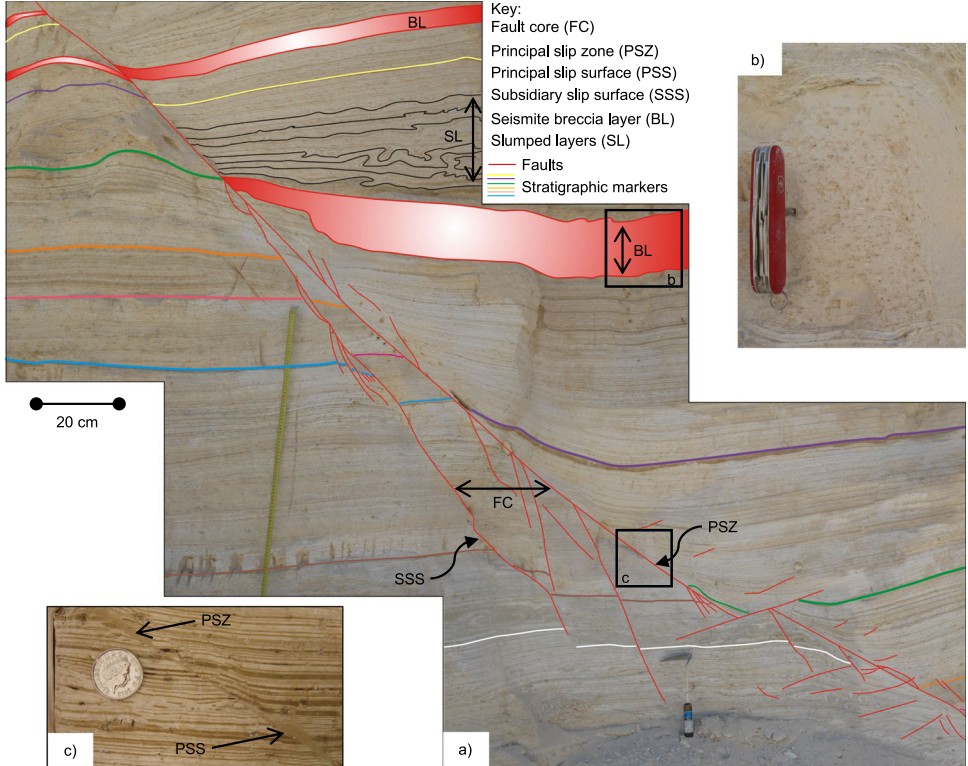

**Fig. 3 | Syn-sedimentary co-seismic faults in the MFZ. a** Image showing the structure of syn-sedimentary fault LIF2.4 (max displacement $d = 337$ cm) in the MFZ, including sedimentary (coloured lines), mixed (ML highlighted in red) and slumped layers (SL highlighted in black). Principal slip zone (PSZ) and subsidiary slip surfaces (SSS) in the fault core (FC) are also highlighted. **b** Image showing a mixed layer seismite, which comprises very fine-grained re-deposited silt. **c** Image showing shear localisation in the fault core (FC), within the principal slip zone (PSZ), bounded on one side by a sharp principal slip surface (PSS).

cataclastic processes (Fig. 7b–d). In the main PSZ, the brine-saturated gouges have a comparable grain size to the initial, undeformed material (Fig. 7b–d). Many of the aragonite crystals have retained their angular acicular shapes and, in a few places, their delicate rosette-type structures are also preserved (Fig. 7c, d). Acicular aragonite crystals are imbricated and aligned with their long axes inclined towards the direction of shear (Fig. 7b). As in the naturally deformed PSZs, these microstructural observations on experimentally deformed gouges suggest that, under in situ brine-saturated conditions, the dominant co-seismic deformation mechanism was particulate flow. Conversely, the simulated gouges sheared under room-humidity conditions show evidence of intense grain size reduction by fracturing and cataclastic processes in the PSZ, where sub-rounded clasts of various sizes are observed (Supplementary Fig. 8), as opposed to the elongated, acicular clast shapes observed in the brine-saturated sheared gouges.

### The energy balance of near-surface ruptures in soft sediments

The total energy released during an earthquake is partitioned into three primary components: fracture energy ($E_G$), dissipated by on- and off-fault damage processes; frictional heat ($E_H$), dissipated along the fault interface; and radiated energy ($E_R$), emitted as potentially hazardous seismic waves[8]. The main components of the earthquake energy balance are graphically represented in Fig. 8a by the areas of the trapezoidal surface below a slip-weakening curve. The total mechanical, or fracture, energy ($E_G$) dissipated by fault processes during rupture propagation is equal to the sum of the breakdown work done on-fault, within the PSZs ($E_G^{PSZ}$) of the fault core, and off-fault ($E_G^{Off\text{-}fault}$), in the surrounding damage zone[8–11] (Fig. 8c).

We use our experimental results and field measurements to estimate the on-fault ($E_G^{PSZ}$) and off-fault ($E_G^{Off\text{-}fault}$) breakdown work dissipated along Fault LIF1.1 during a single seismic slip event. Fault LIF1.1 accommodates 1.85 m of localised displacement along the PSZ and

1.86 m of distributed displacement across a 16 m-wide damage zone (Fig. 5a, b; Supplementary Table 2). The preservation of three distinct seismite layers, confidently attributed to three seismic slip events along the fault[18], provides evidence of repeated fault activity. High-resolution, sub-centimetre-scale stratigraphy[18], combined with conventional palaeoseismic methods[18], identified three seismic slip events along LIF1.1 over a ~2 kyr period, each recording ~40–60 cm of co-seismic slip. The most recent event occurred ~25 kyr ago[18]. Based on this, the $E_G^{Off\text{-}fault}$ derived from the transect is divided by three to estimate the average off-fault energy dissipated per seismic event, corresponding to ~0.62 m of slip along the PSZ, consistent with an earthquake of approximately $M = 6$[31].

The slip-weakening curves produced during our high-velocity experiments on the brine-saturated Lisan Formation gouges for $\sigma_n = 9$ MPa (Fig. 7a) have been used to calculate $E_G^{PSZ}$ for Fault LIF1.1 (e.g., Fig. 8b; see 'Methods'–'Estimation of fracture energy'). Our calculations yield values of $E_G^{PSZ} = 0.16$ MJ m$^{-2}$ per event of 0.62 m of slip, to a corresponding depth of <1 km (Fig. 9a). The small fracture energy values obtained point to low energy dissipation in the PSZ during co-seismic slip. This result agrees with the minimal grain-scale fracturing observed in the natural (Fig. 4) and experimentally deformed gouges (Fig. 7b–d) at co-seismic conditions.

Meso-scale off-fault deformation cannot be reproduced in these small-scale laboratory experiments. Hence, when only the experimental viewpoint is considered, the total fracture energy is underestimated as off-fault damage is not accounted for in the energy balance (e.g., $E_G^{Off\text{-}fault} = 0$; Fig. 8b), yielding $E_G = E_G^{PSZ} = 0.16$ MJ m$^{-2}$ (Figs. 8b and 9a). The fracture energy values determined for fault LIF1.1 are approximately an order of magnitude lower than seismological estimates, which are consistently reported in previous studies[9,41,46] to average about 1–2 MJ m$^{-2}$ for natural earthquakes exhibiting comparable slip ($\approx0.6$ m) and magnitude ($M_w \approx 6$).

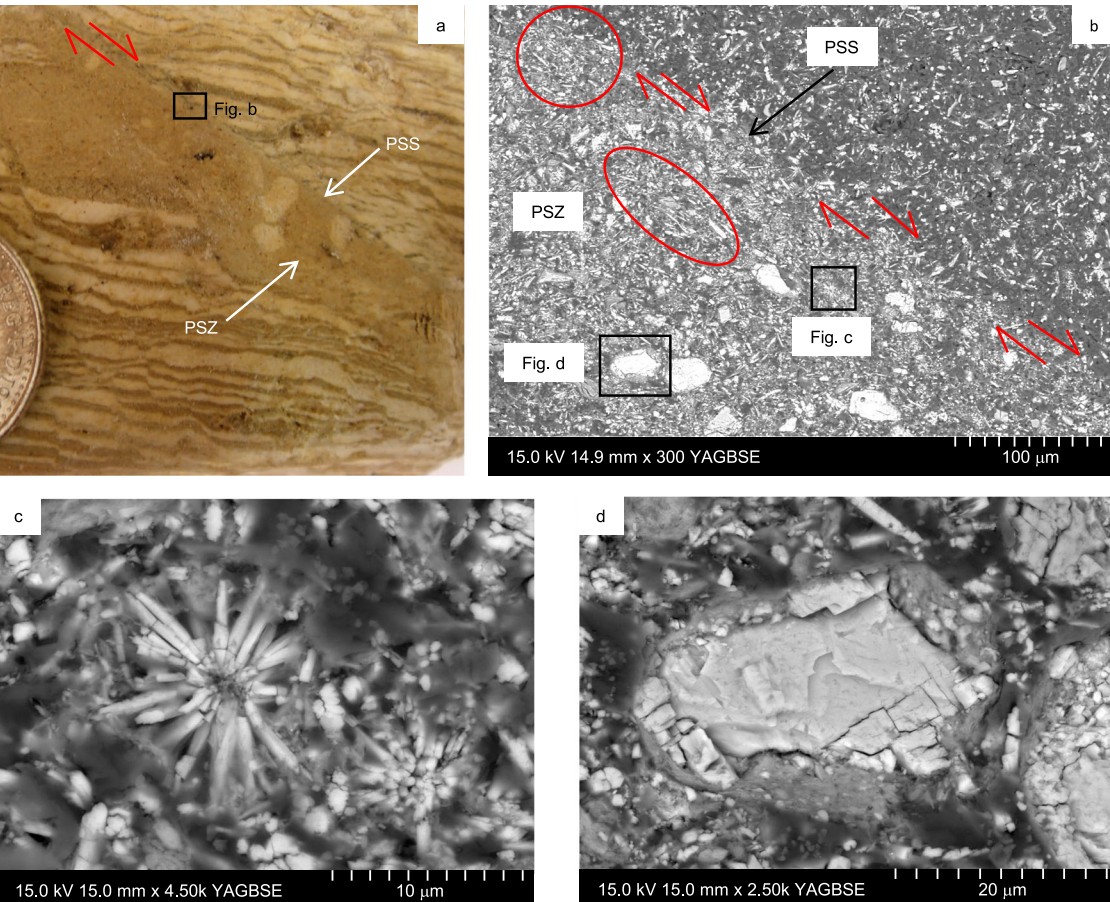

**Fig. 4 | Microstructures of the natural co-seismic principal slip zone (PSZ). a** PSZ of a fault with maximum observed displacement of ~175 cm, with widths up to 8 mm. The contacts (principal slip surface, PSS) between the PSZ and host rock are very sharp. **b** Back Scatter Electron (BSE) image of the PSZ and principal slip surface PSS shown in **a**. Zones highlighted by red circles show the preferential alignment of acicular aragonite clasts in the PSZ, parallel to the PSS. **c** Close-up of intact aragonite rosette within the PSZ shown in (**b**), which also has a cortex of clay starting to form around it. **d** Close-up image of a clast-cortex aggregate in the PSZ shown in (**b**), which suggests rolling of clasts, in line with a particulate flow mechanism of grain-scale deformation.

A more realistic estimation of the dissipated fracture energy for fault LIF1.1 can be obtained if the contribution to fracture energy of meso-scale off-fault deformation ($E_G^{\text{Off-fault}} \neq 0$) seen in the field is also considered (e.g., Fig. 8c). $E_G^{\text{Off-fault}}$ is estimated by calculating the energy required to create hybrid extensional/shear fractures and disaggregation bands within the damage zone (Fig. 6; see 'Methods'–'Estimation of fracture energy'). Using the structural data collected along a transect in the 16 m wide damage zone of fault LIF1.1 (Fig. 5, Supplementary Table 2), we calculated that $E_G^{\text{Off-fault}}$ ranges between about 0.9 and 1.5 MJ m$^{-2}$ (Fig. 9a). The range of values refers to the assumed end-member scenarios of no healing ($E_G^{\text{Off-fault}} = 0.9$ MJ m$^{-2}$) and healing ($E_G^{\text{Off-fault}} = 1.5$ MJ m$^{-2}$) of structures occurring during the interseismic period (Fig. 9a; 'Methods'–'Estimation of fracture energy'). This yields a total fracture energy $E_G = (E_G^{\text{PSZ}} + E_G^{\text{Off-fault}}) = 1.06\text{–}1.66$ MJ m$^{-2}$ (Fig. 9a).

To conclude, when off-fault damage is taken into account, the total fracture energy values obtained for the near-surface co-seismic fault LIF1.1 are comparable to average seismological estimates (1–2 MJ m$^{-2}$, averaged over the full fault depth) and fall within the broad scatter observed in geological and modelling estimates for natural events with similar slip (≈0.6 m) and magnitude ($M_w \approx 6$)[9,41,46] (Fig. 9b).

### Energy dissipation in shallow rupture: the role of off-fault damage

To illustrate the general relationship between stress, fracture energy dissipation, and rupture propagation, we employ an analytical solution for a simplified in-plane shear (mode II) rupture propagating at sub-Rayleigh velocity ($v_r < c_r$) along a planar fault with spatially variable initial stress, where $v_r$ is the rupture velocity and $c_r$ the Rayleigh wave speed (see 'Methods'–'Modelling of rupture velocity'). While advanced numerical models have incorporated the effects of the free surface and material heterogeneities[47,48], and it is well established that inelastic dissipation, such as off-fault damage or the formation of high-angle secondary fractures, can significantly influence rupture speed[14], our focus here is different. We aim to isolate and examine the specific impact of frictional dissipation, as estimated directly from laboratory measurements and field observations of multiple sub-parallel fault strands. To this end, we adopt a simplified analytical model that incorporates the measured dissipation while intentionally excluding additional complexities.

In our simplified model, we calculate rupture velocity $v_r$ for a rupture nucleating at depth (6.6–10 km) and propagating upward, ultimately entering the shallow, soft sediments (Fig. 10a; see 'Methods'–'Modelling of rupture velocity'). We examine six scenarios, combining three different thicknesses of the shallow layer (300 m, 1.4 km, and 2.7 km; Fig. 10b–d) with three values of dissipated fracture energy ($E_G$).

Cases b and c represent generic configurations with 2.7 km and 1.4 km thick shallow layers, respectively, while case d reflects the specific conditions of the Lisan Formation at the study site, with a ~300 m thick shallow layer. For each thickness, we test three dissipation scenarios: (1) $E_G = 1.66$ MJ m$^{-2}$, equivalent to the fault root, incorporating both on- and off-fault energy dissipation and interseismic

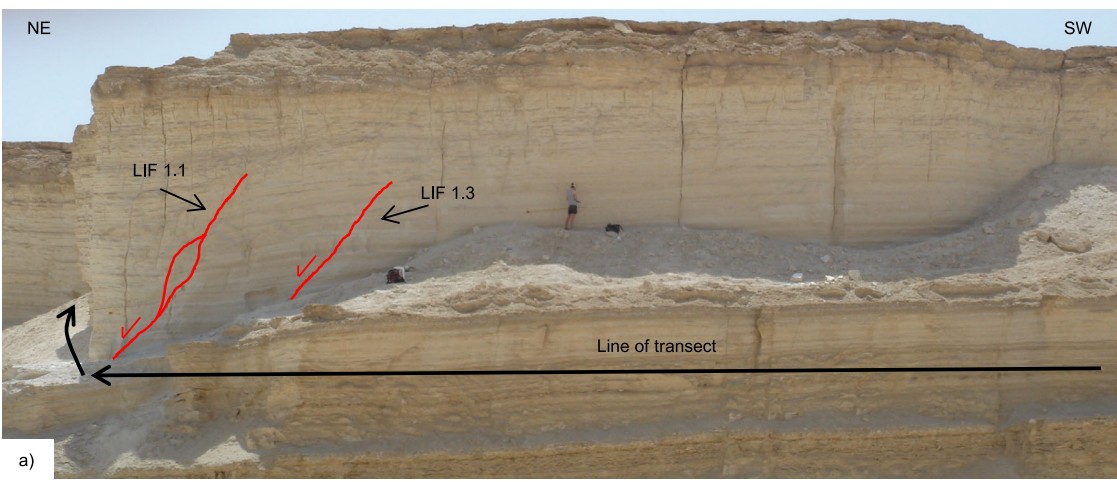

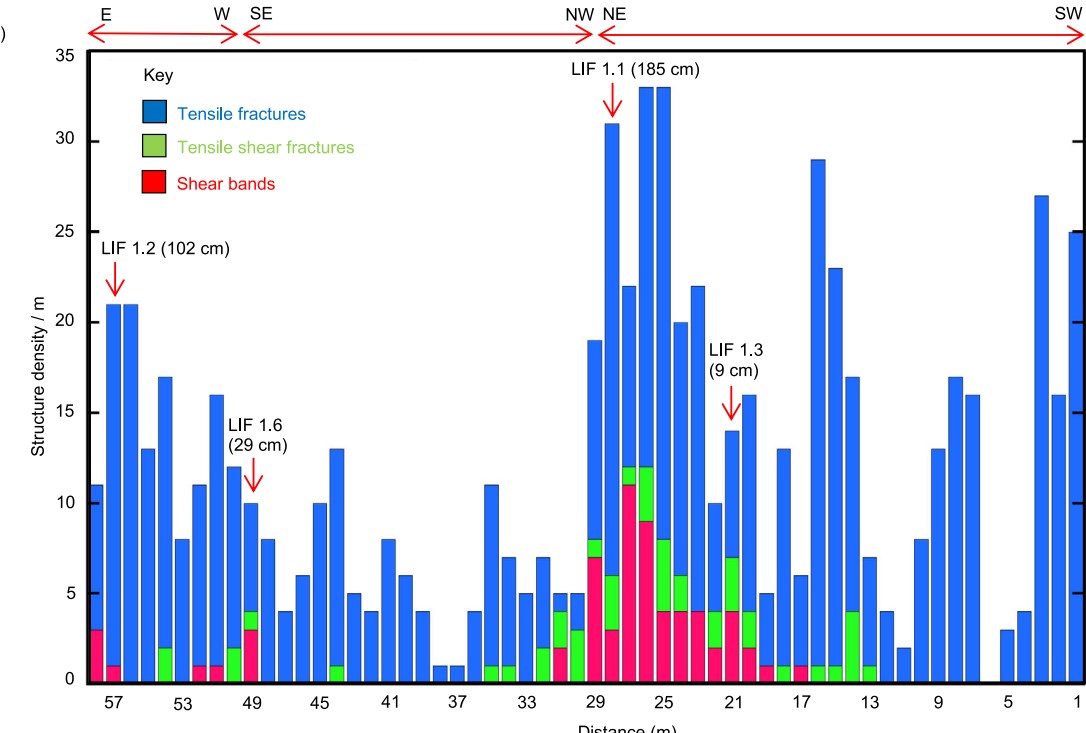

**Fig. 5 | Damage zone of natural co-seismic syn-sedimentary faults. a** Photo of a section of outcrop across which a structural transect was conducted, the results of which are presented in **b**. Faults LIF 1.1 and LIF 1.3 are labelled; faults LIF 1.2 and LIF 1.6, also included in the transect data, are around the corner of the outcrop photographed here, indicated by the curved arrow. Note the presence of sub-vertical tensile fractures, which transect the whole outcrop, whereas faults terminate before the top of the outcrop. **b** Structural transect data that includes four faults, the locations and displacements of which are labelled on the plot. See text for full description.

healing (Fig. 8c); (2) $E_G = 1.06\ \mathrm{MJ\,m^{-2}}$, equivalent to lower dissipation, without interseismic healing, but still accounting for both on- and off-fault processes (Fig. 8c); (3) $E_G = 0.16\ \mathrm{MJ\,m^{-2}}$, equivalent to minimal dissipation, considering only on-fault fracture energy within the PSZ (Fig. 8b). Our preferred interpretation for Lisan co-seismic faulting corresponds to case d with $E_G = 1.66\ \mathrm{MJ\,m^{-2}}$ or $1.06\ \mathrm{MJ\,m^{-2}}$ (Fig. 10d). In this scenario, rupture propagation is governed by both on- and off-fault dissipation, consistent with field observations and laboratory measurements of frictional energy in brine-saturated Lisan sediments under in situ conditions and seismic slip rates.

Despite the absence of prestress in the shallow layer, rupture propagates into it under the momentum imparted from the fault root. In the low-dissipation case ($E_G = 0.16\ \mathrm{MJ\,m^{-2}}$), the rupture initially accelerates upon entering the shallow layer due to the sudden drop in energy

dissipation, but subsequently decelerates as prestress is lacking and energy flow becomes depleted (Fig. 10b–d). Nevertheless, in this scenario, the rupture consistently reaches the surface at relatively high velocity, regardless of the shallow layer thickness. In contrast, when off-fault damage is included ($E_G = 1.06$ or $1.66\ \mathrm{MJ\,m^{-2}}$), rupture deceleration occurs immediately upon entering the shallow layer. In the highest dissipation scenario ($E_G = 1.66\ \mathrm{MJ\,m^{-2}}$), rupture is arrested at depth when the shallow layer is thick (2.7 km, Fig. 10d), but still reaches the surface, albeit with significantly reduced velocity, in the thinner layers (Fig. 10b, c).

Our results show that a high dissipation, in line with our field estimates ($E_G = 1.66\ \mathrm{MJ\,m^{-2}}$), allows a significant reduction in rupture speed (Fig. 10b, c), or even induces the rupture to stop at depth for a scenario where the layer of soft sediments is 2.7 km thick (Fig. 10d).

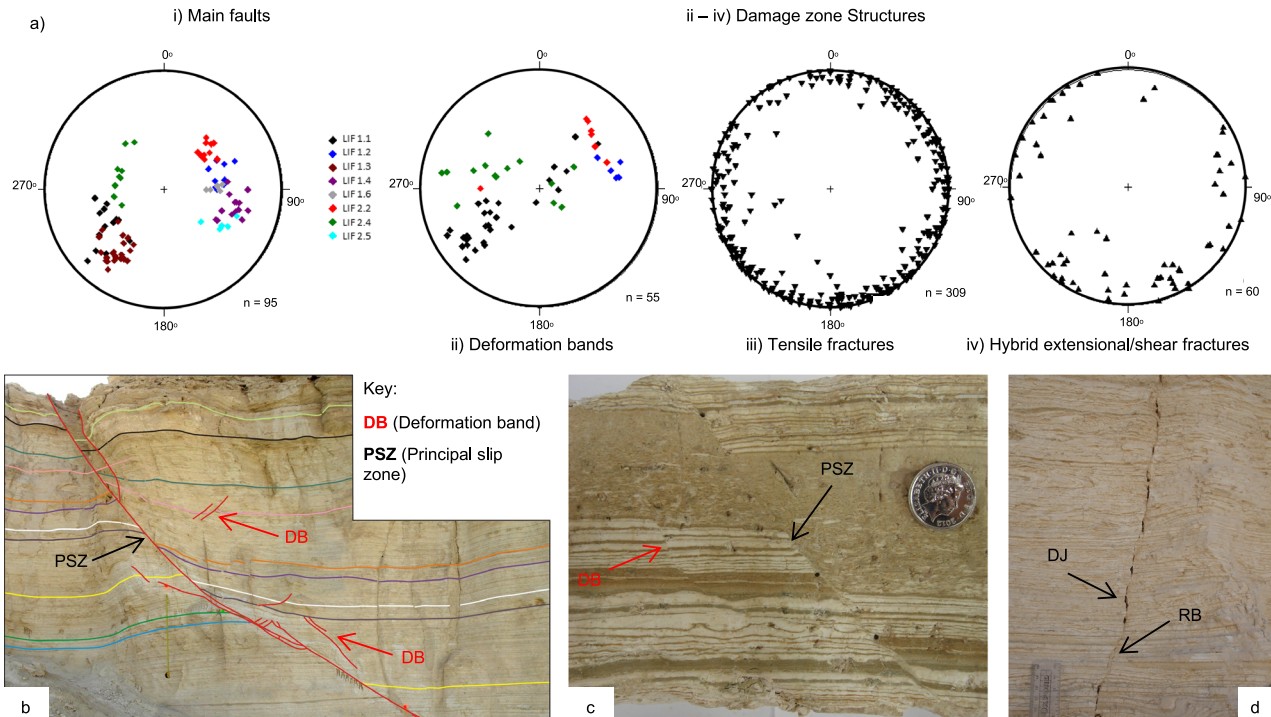

**Fig. 6 | Damage zone shear structures. a** Equal-area, lower hemisphere stereographic projections of: (i) the main faults studied; (ii) deformation bands within the damage zones of the main faults; (iii) tensile fractures not related to the studied faults (see details in the main text); (iv) hybrid extensional-shear fractures within the damage zones of the main faults. Deformation bands (DB) formed in the damage zone of fault LIF 2.4, $d_{max}$ = 337 cm, viewed in outcrop (**b**) and hand specimen **c. d** Hybrid extensional-shear fracture in the damage zone showing extensional (dilational jogs, DJ) and contractional (restraining band, RB) structures.

## Discussion

Our findings shed light on the structure and deformation mechanisms of co-seismic faults in shallow, soft sediments, an environment where energy dissipation during near-surface rupture remains poorly constrained[9].

The MFZs, formed during co-seismic rupture in the clay-rich sediments of the Lisan Formation, show both notable similarities to and key differences from faults formed at depth in strong, low-porosity rocks. The absence of grain-size reduction in the PSZs suggests that cataclastic deformation, a major energy-dissipating mechanism in brittle crustal faults[6,9,49], did not occur. Field and experimental evidence therefore supports the inference that, in brine-saturated, clay-rich soft sediments, little energy is dissipated through the formation of new grain surfaces within the PSZ. The most significant difference lies in the dominant deformation mechanisms: while deep, cohesive, low-porosity rocks deform primarily through fracturing and cataclasis[49], the near-surface, brine-saturated, poorly lithified sediments studied here appear to deform mainly by fluidised granular or particulate flow. These results indicate that under such in situ conditions, on-fault energy dissipation is limited.

Despite this, the MFZ faults show a well-defined fault core–damage zone structure, with fracture density decreasing away from the fault core, patterns consistent with deeper seismogenic faults[39,40]. Furthermore, the widths of damage zones, fault cores and PSZs increase with displacement, following trends typical of fault zones in the brittle crust[39,40]. These observations imply that, despite differences in lithology and mechanical strength, shallow faults in soft sediments can exhibit structural and scaling behaviours comparable to those of deeper crustal faults[39].

To quantify fracture energy dissipation during rupture, we integrated field observations with mechanical data from the MFZ. Our estimates (~1–2 MJ m$^{-2}$) align with geological, seismological, and

modelling-derived values for fracture energy in natural earthquakes of similar magnitude (Fig. 9b). These results suggest that energy dissipation during near-surface rupture in soft sediments is governed by the balance between minimal on-fault dissipation in the lubricated PSZ and significant off-fault dissipation through distributed shear in the broad damage zones. To a first approximation, our results align with previous studies that modelled the impact of co-seismic off-fault fracturing on the overall earthquake energy budget in deeper, mechanically stronger rocks[11,14]. For example, Okubo et al.[14] found that up to half of the fracture energy of the main fault can be dissipated in the surrounding medium through tensile, shear, and hybrid fractures, structures also observed in the damage zones of the MFZ (Fig. 5b).

Building on these observations, we used a simplified model to explore the conditions under which rupture propagates, or stalls, near the surface, particularly in fault segments with low initial stress, as is characteristic of the MFZ. We intentionally excluded complexities such as elastic heterogeneity, density variations, and free-surface effects, which have been addressed elsewhere[14,47,48]. Previous studies have attributed shallow rupture velocity to low initial stress, reduced seismic wave speeds in soft sediments[50], and velocity-strengthening behaviour of clay-rich materials[51]. Our results expand on this by explicitly testing the effect of fracture energy $E_G$ on rupture velocity, as described in Eqs. 22–24 ('Methods'−'Modelling of rupture velocity'). While often acknowledged in theory, the role of fracture energy is frequently overlooked due to the difficulty of quantifying off-fault energy dissipation in natural settings[9,11].

Our analysis demonstrates that incorporating off-fault fracture energy dissipation significantly decelerates or even arrests rupture in the shallow fault segment (Fig. 10b). This behaviour requires that off-fault energy dissipation remains effective even under low normal stress, as observed in the MFZ. Similar to our findings in

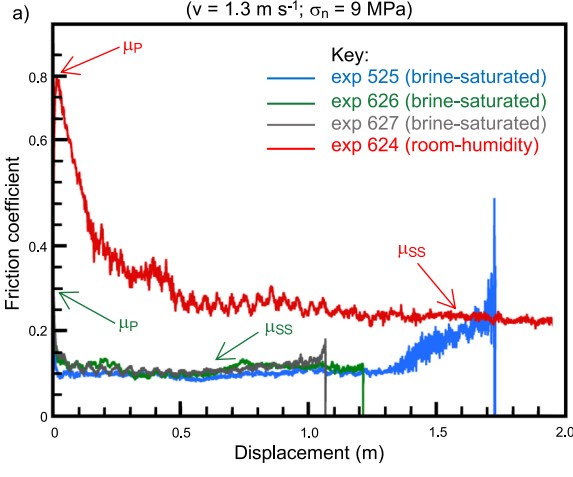

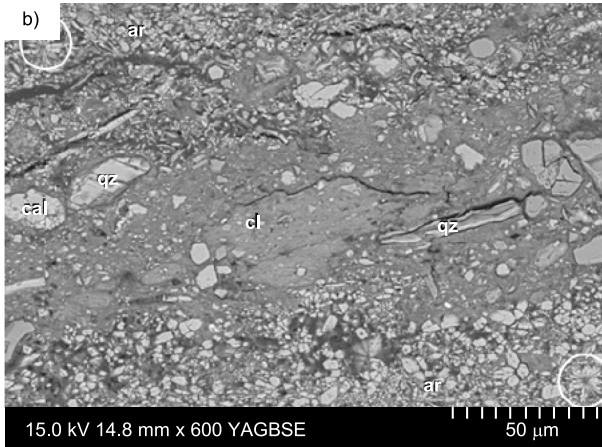

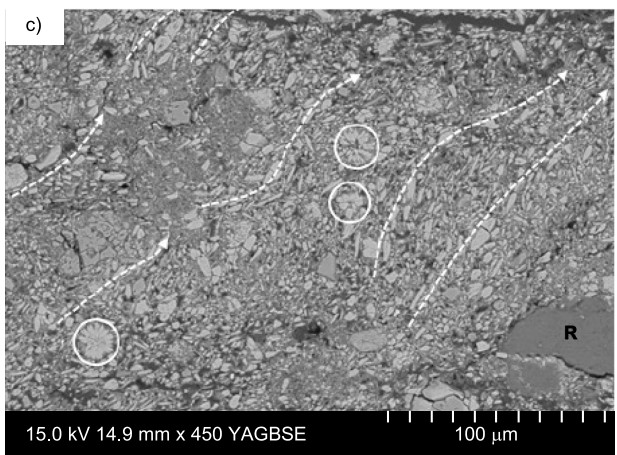

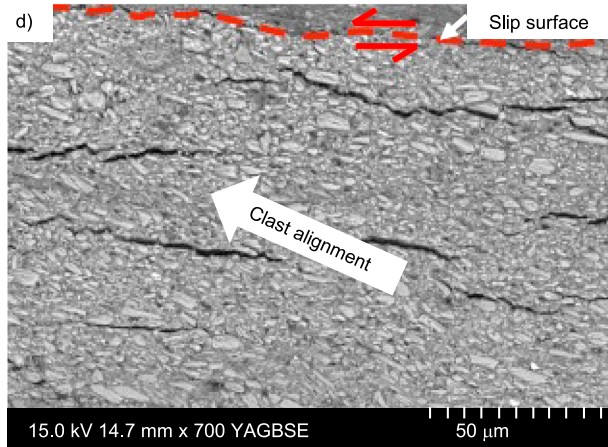

**Fig. 7 | Friction evolution, microstructure and deformation mechanisms of experimental seismic principal slip zone. a** Friction evolution of room-humidity and brine-saturated gouges deformed at 1.3 m s⁻¹ under 9 MPa normal stress shows transient weakening, over a slip-weakening distance from peak ($\mu_P$) to steady-state ($\mu_{SS}$) friction. **b** Back Scatter Electron (BSE) image of un-sheared microstructures (post-compaction) of gouges subjected to 18 MPa normal stress. Larger clasts are labelled to aid mineral identification: cal calcite, qz quartz, ar aragonite. Preserved radial arrangements of acicular aragonite crystals are circled. **c** Microstructures of brine-saturated gouges deformed at 1.3 m s⁻¹ under 1 MPa normal stress, showing dominant flow directions highlighted by arrowed lines. Examples of relatively intact radial arrangements of aragonite crystals are circled. **d** Microstructures of brine-saturated gouges deformed at 1.3 m s⁻¹ under 9 MPa normal stress. The arrow indicates the dominant orientation in which elongated acicular aragonite clasts are aligned.

shallow, low-stress settings, Okubo et al.[14] also predicted that in deeper, high-stress environments, off-fault damage can slow rupture propagation, and delay or prevent supershear transition. In both shallow and deep cases, the decelerating effect of off-fault damage on rupture velocity is attributed to an increase in the critical slip distance: more slip is required before the fault weakens sufficiently for efficient propagation under low residual friction (e.g., compare Fig. 8b, c).

Our results show that an increased ratio of dissipated fracture energy $E_G$ to radiated energy $E_R$ yields low radiation efficiency $\eta_R < 1$, where $\eta_R = E_R/(E_R + E_G)$. This is consistent with seismological observations of slow rupture and low $\eta_R$ in the upper crust[3,8], and mirrors predictions for deeper crustal settings, where co-seismic off-fault damage reduces the energy available for seismic radiation[14].

In summary, our findings offer a field- and experiment-based explanation for key geophysical features of shallow ruptures in soft, clay-rich sediments, including slow rupture velocity and low radiation efficiency. These findings have important implications for models of rupture dynamics and for seismic hazard assessments in shallow regions of the crust dominated by poorly lithified sediments.

## Methods
### Estimation of fracture energy
The total amount of the earthquake energy budget dissipated as fracture energy $E_G$ during rupture propagation is equal to the sum

$$E_G = E_G^{\text{PSZ}} + E_G^{\text{off-fault}} \qquad (1)$$

where $E_G^{\text{PSZ}}$ is the breakdown work done within the principal slip zone PSZ and $E_G^{\text{off-fault}}$ is the breakdown work done off-fault in the damage zone[8-10] (see Fig. 8c in the main text).

### Estimate of the principal slip zone breakdown work ($E_G^{\text{PSZ}}$)
The mechanical work dissipated within the PSZ ($E_G^{\text{PSZ}}$) can be expressed in terms of the anelastic strain dissipated per unit fault area, and it is equivalent to the area of the surface under the experimental slip-weakening curves represented by $E_G^{\text{PSZ}}$, as depicted in Fig. 8b in the main text. As a first approximation, $E_G^{\text{PSZ}}$ can be calculated according to equation

$$E_G^{\text{PSZ}} = \frac{1}{2}\left(\left(\tau_p - \tau_r\right) \cdot D_w\right) \qquad (2)$$

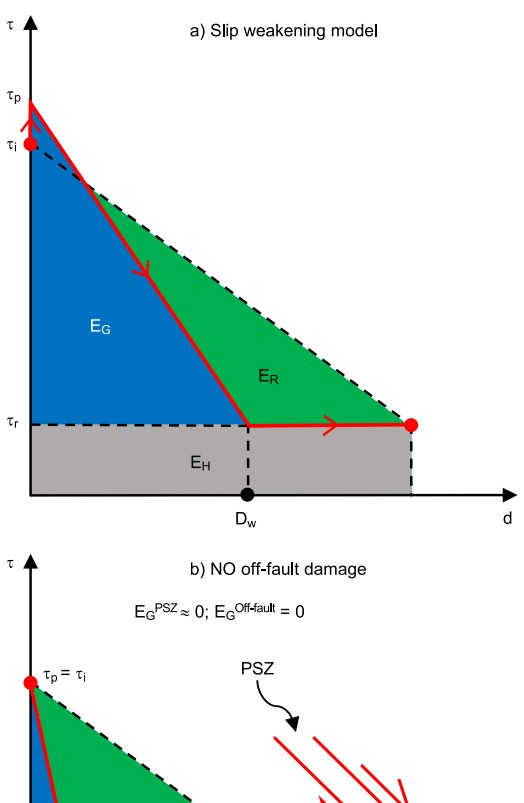

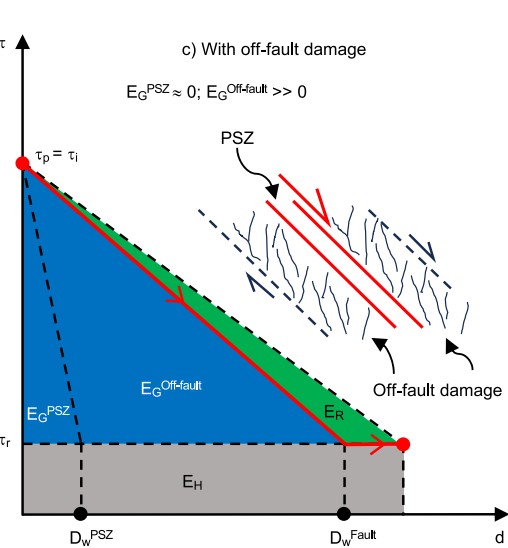

**Fig. 8 | Graphical representations of the energy budget partitioning of an earthquake for different scenarios. a** The partitioning of earthquake energy according to the slip-weakening model[8]. $\tau$ = Shear stress; $d$ = slip; $D_W$ = slip weakening distance; $\tau_p$ = Peak shear stress; $\tau_i$ = initial shear stress; $\tau_r$ = residual shear stress; $E_G$ = fracture energy; $E_R$ = radiated energy; $E_H$ = Frictional heat. **b** The partitioning of energy as implied by the frictional evolution of a typical high-velocity friction experiment. $E_G^{PSZ}$ and $E_G^{Off\text{-}fault}$ are the on-fault and off-fault energy dissipated within the PSZ and damage zone, respectively, during seismic slip. **c** The proposed partitioning of energy during rupture propagation through a natural, fluid-saturated, clay-rich fault zone at shallow depth.

have been calculated using the Coulomb failure criterion

$$\tau_p = C + (\tan \varphi)\sigma_n \tag{3}$$

substituting the values of $C \approx 0$ for the cohesion and $\varphi = 34°$ for the angle of internal friction, obtained from undrained triaxial tests performed on undisturbed samples of brine-saturated Lisan Formation sediment[52]. The other average parameters $\tau_r$ and $D_w$ have been obtained from our high-velocity friction experiments performed on the brine-saturated Lisan Formation sediment at normal stress $\sigma_n = 9$ MPa (Supplementary Table 1).

**Estimate of off-fault breakdown work ($E_G^{off-fault}$)**
The mechanical work dissipated off-fault ($E_G^{off-fault}$) is given by the energy required to create hybrid shear fractures and deformation bands within the damage zone. If we consider a population of $N$ shear structures, let the final slip $d_i$ be the final slip on the $i$th fault branch in the population. Then the total energy $E_D^i$ dissipated on that branch equates to the area of the right trapezoid surface between the slip-weakening curve and residual friction (e.g., $d_i = d_2$, $d_3$ or $d_4$ in Supplementary Fig. 9). Therefore,

$$E_D^i = \int_0^{d_i} \tau(d)dd \tag{4}$$

whereas the total dissipated energy $E_D$ from slip on all N shear structures of the array is

$$E_D = \sum_{i=1}^{N} \int_0^{d_i} \tau(d)dd \tag{5}$$

The total dissipated energy $E_D$ can be partitioned into the sum of residual frictional work $E_H$ and fracture energy $E_G$ (Fig. 8a in the main text) such that

$$E_D = E_G + E_H \tag{6}$$

with $E_H = \tau_r D_T$, where $D_T = \sum_{i=1}^{N} d_i$ is the total cumulative displacement across the shear structure population. Therefore, we may write that fracture energy $E_G^{Off\text{-}fault}$ is equal to

$$E_G^{Off-fault} = E_D - E_H = \sum_{i=1}^{N} \int_0^{d_i} \tau(d)dd - \tau_r \sum_{i=1}^{N} d_i \tag{7}$$

$$E_G^{Off-fault} = \sum_{i=1}^{N} \int_0^{d_i} (\tau(d) - \tau_r)dd \tag{8}$$

And the contribution of each $n$th individual fault is

$$E_G^i = \int_0^{d_i} (\tau(d) - \tau_r)dd \tag{9}$$

where $\tau_p$ is the peak shear stress, $\tau_r$ is the residual shear stress and $D_w$ is the slip weakening distance (see Fig. 8b in the main text). We obtain that $E_G^{PSZ} = 0.16$ MJ m$^{-2}$ for a single seismic event of 0.62 m of slip along the main PSZ, when Eq. 2 is solved using the average values of $\tau_p = 6.07$ MPa, $\tau_r = 0.9$ MPa and $D_w = 0.06$ m for brine-saturated fault gouges. The peak shear stress $\tau_p$ values at which slip initiates in the PSZ

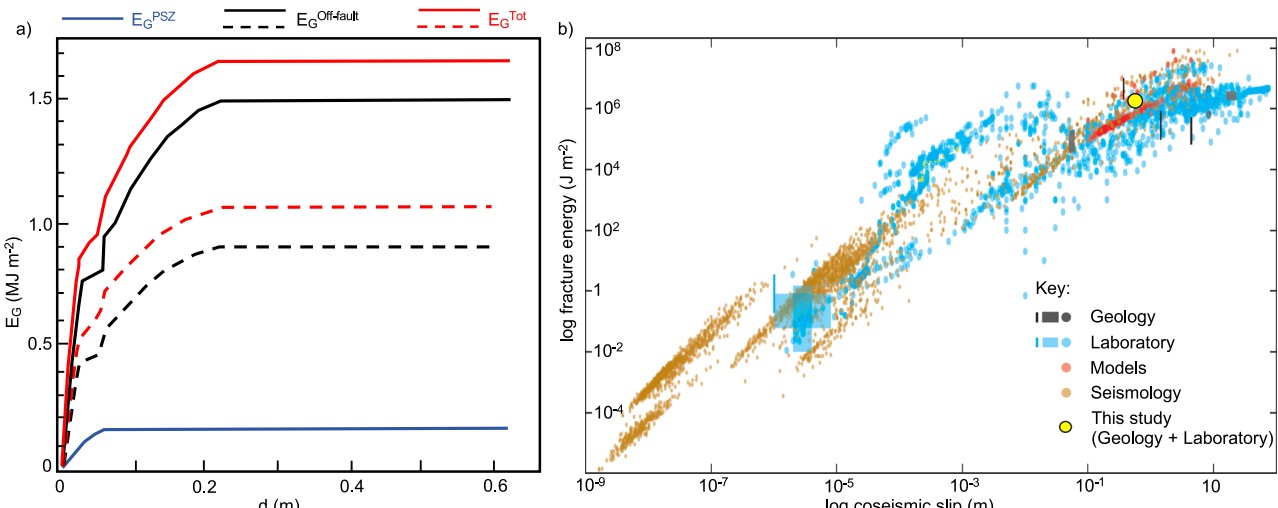

**Fig. 9 | Earthquake energy budget estimated for a single seismic rupture in a natural fault in soft sediments. a** Estimated partitioning of on- and off-fault dissipated energy during rupture propagation through a natural, fluid-saturated, fault zone in soft sediments (LIF1.1). $d$ = Slip; $E_G^{Tot}$ = total fracture energy; $E_G^{PSZ}$ = on-fault energy dissipated within the principal slip zone (PSZ); $E_G^{Off-fault}$ = off-fault energy dissipated within the damage zone. Dashed and solid lines refer to the two end-member scenarios of no healing and healing of structures in the damage zone occurring during the interseismic period, respectively. **b** Scaling of fracture energy densities with slip of rupture propagation obtained from geological, seismological, modelling and laboratory studies[9]. Fracture energy estimates from this study have been added to the plot, showing a good fit with seismological estimates. Adapted from Cocco et al.[9].

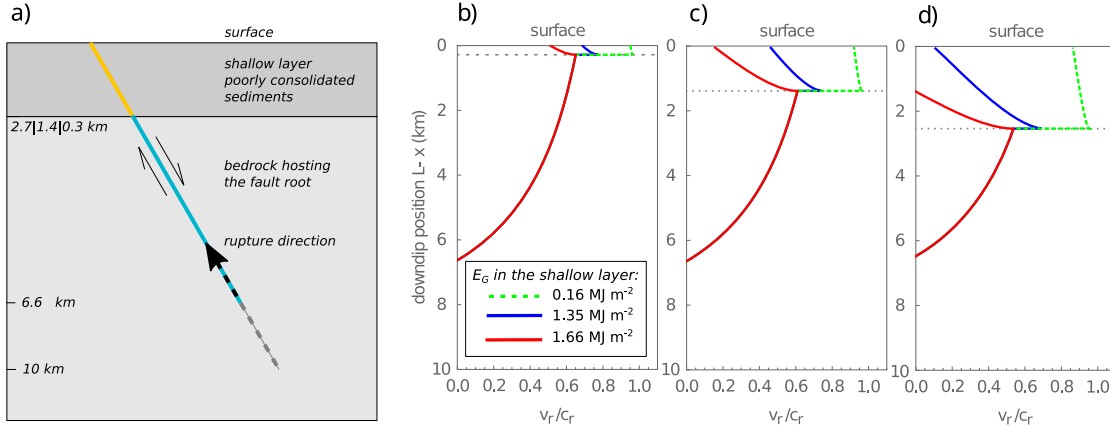

**Fig. 10 | Rupture velocity models during propagation along faults that intersect a shallow layer of soft sediments. a** Idealised model of rupture propagation from a deeper fault root and through a shallow layer of poorly consolidated sediments. The fault root has a prestress 3 MPa and the shallow layer has no prestress. Modelled rupture velocity ($v_r$) evolution (normalised by Rayleigh wave speed, $c_r$) during rupture propagation through shallow layers with 2.7 km (**b**), 1.4 km (**c**) and 0.3 km (**d**) thickness (the latter case corresponds to the approximate thickness of the Lisan formation). In all cases (**b**, **c**, **d**), the root fault section has fracture energy $E_G$ = 1.66 MJ m$^{-2}$. In the shallow fault section, models with three different values of fracture energy were implemented: 0.16 MJ m$^{-2}$, 1.06 MJ m$^{-2}$ and 1.66 MJ m$^{-2}$ (the latter fracture energy values correspond to the frictional dissipation estimated from experimental and field measurements, accounting for both on- and off-fault deformation of the principal fault). See text for further details.

It is noteworthy that, because the secondary faults undergo a modest slip $d_i$ with incomplete weakening, their relatively high sliding friction value further increases their dissipative power (Supplementary Fig. 10). As illustrated in Supplementary Fig. 10, approximating the weakening curve by a linear trend between $\tau_p(d=0)$ and $\tau_r(d \geq D_w)$, we can calculate the fracture energy $E_G^i$ dissipated during the slip of a single shear fracture with an amount of slip $d_i < D_W$ as

$$E_G^i(d_i < D_W) = (\tau_p - \tau_r)\left(d_i - \frac{1}{2}\frac{d_i^2}{D_w}\right) \quad (10)$$

On the other hand, the fracture energy $E_G^i$ dissipated during the slip of a single shear fracture with an amount of slip $d_i \geq D_W$,

is equivalent to the area of the surface under the slip-weakening curve up to $D_W$ (e.g., $d=d_1$ in Supplementary Fig. 9) and can be calculated as

$$E_G(d_i \geq D_w) = \frac{1}{2}(\tau_p - \tau_r)D_w \quad (11)$$

The total fracture energy dissipated during shear of off-fault slip structures can be written as the summation

$$E_G^{Off-fault} = \sum_{i=1}^{N}(\tau_p - \tau_r)\left(d_i - \frac{1}{2}\frac{d_i^2}{D_w}\right) + \sum_{i=m+1}^{N}\frac{1}{2}(\tau_p - \tau_r)D_w \quad (12)$$

where $N$ is the total number of shear structures created in the damage zone of a fault during a single seismic event, $m$ is the number of structures with an amount of slip $d < D_W$, $\tau_p$ is the peak shear stress at which a new fault forms in intact rocks or a pre-existing one is reactivated, $\tau_r$ is the residual shear stress on the fault upon attainment of the slip weakening distance $D_W$.

The data collected from a structural transect along the fault LIF1.1 in the Lisan formation (Supplementary Table 2) were used in Eqs. 11 and 12 to calculate its total $E_G^{\text{off}-\text{fault}}$. In our calculations, we consider two end-member scenarios where the structures in the surrounding damage zone, outwith the PSZ, formed by either failure of intact brine-saturated Lisan Formation sediment or reactivation of pre-existing structures, which were created during one of the 3 distinct seismic events of similar magnitude. For the newly formed faults scenario, the value $\tau_p = 6.07$ MPa was used, obtained from Eq. 3 for $C = 0$, $\varphi = 34°$ and $\sigma_n = 9$ MPa, consistent with the normal load applied during our high-velocity friction experiments. For the reactivated faults scenario, $\tau_p = 2.97$ MPa was used, obtained from our high-velocity friction experiments. For both scenarios, the values $\tau_r = 0.9$ MPa, $D_w = 0.06$ m and $\tau_d = $ to the specific shear stress value attained after a final amount of slip d on the fault (e.g., Supplementary Fig. 10) were used, all values obtained from our high-velocity friction experiments on brine-saturated Lisan Formation sediment performed at $\sigma_n = 9$ MPa. When these values are used to solve Eqs. 11 and 12, and after dividing the value obtained for a factor of 3, we obtain values of the total off-fault fracture energy $E_G^{\text{off}-\text{fault}} = 0.9$–1.5 MJ m$^{-2}$ for reactivated and for newly formed ones, respectively.

## Modelling of rupture velocity

To illustrate the general relationship between stress, fracture energy dissipation, and rupture propagation, we use an analytical solution for a simplified in-plane (mode II) rupture model. To a first-order approximation, the Lisan units are assumed as rigid-elastic bodies prior to brittle failure in the low-stress regime, as they show reduced failure threshold imposed by small confining pressure and cohesion[52]. This model describes rupture propagation at sub-Rayleigh velocities ($v_r < c_r$) along a planar fault with spatially variable initial stress, where $v_r$ is the rupture velocity and $c_r$ is the Rayleigh wave speed. The evolution of $v_r$ is calculated based solely on the energy release rate and the dissipated fracture energy. To better understand the role of fracture energy dissipation, particularly as estimated from the Lisan fault case study, we neglect other complexities, such as variations in elastic properties, density, or free-surface interactions. Instead, we employ a simple analytical formulation, which we implement across four distinct scenarios (see details in the Results section of the main text).

The stress intensity factor $K_0$ at time $t$ can be written as[53]

$$K_0(t) = K_0(L) = \frac{\sqrt{\pi}}{2} \int_0^L \frac{\Delta\tau(x)}{\sqrt{L(t) - x}} dx \tag{13}$$

where $L(t)$, is the current rupture tip position at time $t$, and $\Delta\tau(x)$ is the local stress drop at position $x$ that is the location along the fault trace. Assuming that $x$ is the up-dip position along the fault, and using piecewise constant prestress ($\Delta\tau$ for the fault root from 0 to $L_1$, $\Delta\tau_1$ for the shallow part from $L_1$ to $L$) we can write

$$K_0(L) = \Delta\tau\sqrt{\pi L}\left(1 - \left(1 - \frac{\Delta\tau_1}{\Delta\tau}\right)\sqrt{1 - \frac{L_1}{L}}\right) \text{(for } L > L_1) \tag{14}$$

$$K_0(L) = \Delta\tau\sqrt{\pi L} \text{ (for } L \leq L_1) \tag{15}$$

From the stress intensity factor, we can compute[54] the energy release rate $G_O$ for a crack propagating at vanishing slow velocity in

plane-strain conditions as

$$G_0(L) = \frac{K_0^2(L)}{2\mu'}(1 - \nu) \tag{16}$$

where $\mu'$ is the shear modulus, $\nu$ is the Poisson ratio. For a rupture propagating at sub-Rayleigh velocity $v_r$, the dynamic stress intensity and the dynamic energy release rate for sub-shear ruptures are obtained by adding velocity-dependent functions[52]

$$\begin{aligned} K(L) &= K_0(L)g_1(v_r) \\ G(L) &= G_0(L)g_2(v_r) \end{aligned} \tag{17}$$

For practical purposes, in the subsonic regime, the combined velocity-dependence $g(v_r) = g_1^2(v_r)g_2(v_r)$ can be approximated by $g(v_r) \approx \left(1 - \frac{v_r}{c_r}\right)$ for an in-plane (mode II) crack, to obtain

$$G(L) \approx G_0\left(1 - \frac{v_r}{c_r}\right) \tag{18}$$

where $c_r$ is the Rayleigh wave velocity. For dynamic rupture to be sustained, the energy flow $G$ (Eq. 18) must equal the fracture energy $G_c$ dissipated at the crack tip in the formation of new fault surface, as given by

$$G(L) = G_c(L) \text{ for } G_0 > G_c \tag{19}$$

The rupture velocity $v_r$ can be obtained by equating (Eq. 18), the energy balance between the energy flow $G$ at the crack tip, to the critical energy release rate $G_c$, representing the energy dissipated at the crack tip, and generally considered as a material property[55], such that:

$$v_r = 0 \text{ for } G_0 \leq G_c \tag{20}$$

(note that this corresponds to the Griffith criterion in the limit $v_r \rightarrow 0$).

As proposed in previous studies[9], we assimilate $G_c$ to the breakdown energy $E_G$ (or work of frictional weakening) taking place during the initial slip on the fault. Here, we additionally account for $E_G^{\text{Off}-\text{fault}}$, the work of inelastic deformation distributed within the damage zone, as a contribution to the total dissipated energy, yielding $E_G = E_G^{\text{PSZ}} + E_G^{\text{Off}-\text{fault}}$. The off-fault energy dissipation is not negligible; it takes place during rupture propagation and hence, should be accounted for in the energy balance as discussed elsewhere[10,11,14,41]. Therefore, we posit that $G_c = E_G$—both $E_G^{\text{Off}-\text{fault}}$ and $E_G^{\text{PSZ}}$ can be estimated from field and experimental measurements as illustrated in the main text.

Substituting $G_0$ and $G_c$ we can retrieve the value of rupture velocity by solving the equation

$$G_0(L)g(v_r) = E_G(L) \tag{21}$$

to obtain

$$v_r(L) \approx c_r\left(1 - \frac{E_G(L)}{G_0(L)}\right) \text{ for } E_G > G_0 \tag{22}$$

$$v_r = 0 \text{ for } E_G \leq G_0 \tag{23}$$

with $G_0$ obtained from Eqs. (14) and (16) for the case where $\Delta\tau_1 = 0$ in the superficial part of the fault, such that

$$G_0(L) = \frac{\pi(1 - \nu)}{2\mu}L\left(1 - \sqrt{1 - \frac{L_1}{L}}\right)^2\Delta\tau \tag{24}$$

The equations derived above were used to calculate the rupture velocity $v_r$, for a rupture initiated within a nucleation patch located at the deeper part of the fault (6.6–10 km depth). The rupture then propagates upward toward the shallow soft sediments near the surface. The results are summarised in Fig. 10 and discussed in detail in the Results section of the main text.

## Data availability

The experimental dataset generated in this study and presented in Fig. 7 of the main text and Supplementary Fig. 9 has been deposited in the Zenodo repository and will be accessible at https://doi.org/10.5281/zenodo.17380400. The processed main parameters of the experimental data are provided in Supplementary Table 1 within the Supplementary Information file. The field dataset used to estimate off-fault fracture energy values shown in Fig. 9 is available in Supplementary Table 2 of the Supplementary Information file.

## Code availability

The Wolfram Mathematica code (including usage instructions) used to generate the data plotted in Fig. 10 has been deposited in a public GitHub repository and archived on Zenodo. It is accessible at https://doi.org/10.5281/zenodo.17350858.

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

## Acknowledgements
The authors thank L. Bowen (G. J. Russel microscopy facilities at Durham University) for assistance with the acquisition of SEM images and discussion. This work was funded by the Natural Environment Research Council through a NERC PhD studentship NE/J500215/1 awarded to R.J.B. and a NERC standard grant NE/H021744/1 awarded to N.D.P. S.M. was supported by an ISF grant no. 1164/23.

## Author contributions
N.D.P. conceptualised the idea with the help of S.N. and R.J.B. R.J.B. performed fieldwork, experimental work and microstructure analysis with the help of N.D.P., R.E.H. and S.M. S.N. performed the modelling of results with conceptual inputs from N.D.P. All authors contributed to data analysis, discussion, and interpretation of results. N.D.P. wrote the manuscript with the help of S.N., R.E.H., R.J.B. and S.M.

## Competing interests
The authors declare no competing interests.
