## [Transparent Peer Review file · Nature Communications]

Off-fault damage controls near-surface rupture behaviour in soft sediments

Corresponding Author: Professor Nicola De Paola

Version 0:

Reviewer comments:

Reviewer #1

(Remarks to the Author)

This is a really great paper, well written, with conclusions well supported by the data and modeling. I love the combination of field measurements, lab experiments and modeling to address the contribution to the energy budget from off-fault damage. I found I did not need to edit the paper at all, just noting in 2 spots where a reference might be added.

For a long time it seemed like workers in this field assumed fracture energy was effectively negligible with most/all non-radiated energy being converted to heat. In fact, there is very little, if any, evidence in nature for this assumption about heat. In more recent times, evidence is growing that fracture energy is quite significant when including off-fault damage. My research group is tackling this topic in systems exhumed from brittle-ductile transition depths, and it is really nice to see this study of the near-surface problem. I love the field photos - clearly an excellent place to do this kind of work. I am not surprised to see the large off-fault energy sink in these soft sediments, and I think the numerical simulations predicting that such large off-fault energy dissipation can slow down and arrest ruptures in soft sediments are sound.

I recommend publication in Nature Communications with minimal revision. This is an important topic, and this paper makes an excellent contribution.

Scott Johnson, 11/22/24

(Remarks on code availability)

Reviewer #2

(Remarks to the Author)

The authors study the seismic behaviour of a layer of soft and poorly lithified sediments deposited near the Earth's surface. This scientific question is important to better understand and assess the risk of surface-rupturing earthquakes, which are still poorly understood. To do so, the authors studied an exhumed fault zone along the Dead Sea transform, referred to as the Masada Fault Zone (MFZ). The MFZ is particularly interesting as it cuts through a layer of poorly consolidated sediments deposited at the bottom of the Lake Lisan (the precursor of the current Dead Sea). The authors conducted field work to study the structure of fault zone in this sedimentary Lisan Formation (LF). An important observation reported by this study is that even the material within the principal slip zone (PSZ) shows little grain abrasion and fracturing, which suggests that the seismic deformation mostly arises through grain-scale particulate flow. To test this observation, the authors performed frictional experiments at seismic velocity conditions on sediments of the LF. They showed that the saturation of the sediments with brine is essential to reproduce the deformation and microstructure observed in the field. An important consequence on this particulate flow mechanism is the associated low value of residual friction and fracture energy measured during the experiments. To estimate the total energy dissipated during earthquake rupture, the authors also performed a survey of the damage observed in the field along a transect surrounding the PSZ, from which they estimated the amount of off-fault energy dissipated during the three major seismic events that arose along the MFZ. The comparison with the energy dissipated inside the PSZ (measured in the lab experiments) reveals that co-seismic fracture energy in these soft sediments is mainly due to off-fault dissipation (up to 85%). The paper concludes by presenting results from a model of

shear fracture used to demonstrate the effect of larger off-fault fracture energy on the crack speed.

The research question investigated in the paper is important and the authors conducted convincing field and laboratory works to present original results on the seismic behaviour of soft and poorly consolidated sediments deposited near the surface of the MFZ. The deformation mechanism within the PSZ and the very large portion of off-fault damage are all important observations, different from what is expected for fault zone across more competent rock. In my opinion, the present study could be an important contribution to the field.

However, the discussion section of the paper needs some major transformation. At the moment, it is mostly dedicated to present results from a fracture mechanics model rather than summarizing these important observations made from the field and lab works, and discussing them in the context of previously published works. The proposed model is not well-suited to study the problem of shallow layer of soft sediments and takes the place of important discussion of the original contribution of the paper. For these reasons, I recommend a major revision of the manuscript, particularly its discussion section.

Major points:

- In the discussion, the authors present results based on the model of Fossum and Freund 1973. This fracture mechanics model is derived using boundary conditions and assumptions that are in contradiction with the set-up of interest shown in Figure 10. First the model assumes an unbounded solid and cannot capture the effect of the free surface, which is important in the dynamics of surface-rupturing faults. Second, it assumes homogeneous bulk properties and cannot capture the strong contrast of material properties that exists between the layer of sediments and the bedrock. It seems, for instance, highly unlikely that the rupture will propagate across the sediments at speed close to the Rayleigh wave speed of the bedrock as shown in Figure 10. Third, the model assumes linear elastic fracture mechanics conditions with questionable applicability to a material "which only allows support of very limited elastic stress before failure or ductile deformation" (quoting here lines 270-271 of the paper). I am not against the simplicity of the model, but the conclusions drawn from this model should be taken with much more care. In this context, presenting it as a "quantitative framework for shallow ruptures propagating through soft sediments" is a clear overstatement that should be nuanced.

- Another issue with the model is that it is introduced in the discussion section, which is not the appropriate place to present model and results. Consequently, the paper missed a proper discussion of the original and important results from lab and field works in the context of previous works. How do the measured on-fault and off-fault dissipations compare with other field/lab measurements made on more competent fault rock? How does the portion of off-fault dissipation (85%) compare with values typically assumed/observed in the literature? These are few examples of important aspects that must be discussed in this section of the paper. In my opinion, the fracture mechanics model can simply be removed from the main text. If its purpose is to illustrate the relation between stress, off-fault dissipation and rupture propagation, it should rather be substituted by a discussion of published modelling results, such as the one of Okubo et al. 2019 that quantified how off-fault damage slows down rupture and reduces the radiation efficiency.

Minor points:

As Nature Communications is a multidisciplinary journal, more effort could be made to clarify some concepts and terminology in the paper (see also the line-by-line comments hereafter). For example, the sketch of Figure 10a could be already included in Figure 1 to provide a generic illustration of a surface-rupturing fault crossing a layer of sediments. The energy budget of earthquakes is another central aspect of the article that deserves further precision. For instance, I would recommend clarifying the physical meaning of these different energy contributions that are listed at lines 200-201 (for example: "(...) into the fracture energy E_G that describes the dissipated energy required to bring fault friction down to its residual value, the energy E_H dissipated by slip at residual friction mainly in the form of heat and the energy radiated by seismic waves E_R ").

Line by line comments and suggestions:

Abstract line 18-20: "Our numerical simulations predict that such high amount of off-fault energy dissipation in the shallow portion of the fault can slow down and arrest ruptures in soft sediments." This is an example of claim based on the fracture mechanics model that should be toned down or removed. This is also the only place of the paper that presents this model as "numerical simulations".

Line 29: "The total potential energy of an earthquake" → "The potential energy released during an earthquake"

Line 45-46: "(e.g., energy sinks related to on-fault vs. off-fault deformation)" It is not clear, especially for a broad audience, what "energy sinks" is referring to in this sentence.

Line 109-111: Is the energy dissipated along subsidiary slip surfaces accounted for in the in-fault or in the off-fault fracture energy?

Line 119-121: "It is particularly striking that even within the PSZ, some of the aragonite crystals retain their delicate radial, rosette-like arrangements of needles during deformation" Very nice indeed! Could this indicate that the particulate flow has a very localized flow profile, where most of the flow experiences little shear?

Line 170-171: "These experiments allow us to assess how sensitive co-seismic fault strength is to burial depth in the uppermost part of the crust." From your experiments it seems that brine saturation has a much stronger impact than burial

depth. This is an interesting aspect that is worth highlighting and developing further in the discussion section.

Line 203-206: "The total mechanical, or fracture, energy (E_G) dissipated by fault processes during rupture propagation is equal to the sum of the mechanical work done on-fault, within the PSZs (E_G^{PSZ}) of the fault core, and off-fault (E_G^{Off}), in the surrounding damage zone." This is unclear, because the work of residual friction associated to heat dissipation also corresponds to "mechanical work done on-fault". "Breakdown work" as discussed in Cocco et al. 2023 might be a more precise way to describe and refer to this dissipation.

Line 222: "This result agrees with the minimal grain-scale fracturing observed" It could be insightful to also discuss and compare these measurements of E_G to the fracture energy measured from the experiments at room-humidity that show evidence of fracturing and grain size reduction.

Line 228: "very small, almost negligible when compared to" → "about one order of magnitude smaller than"

Line 230: It would be insightful to compare and discuss the on-fault fracture energy measured in this study with the one reported from the references listed at line 230 in the text, particularly the one measured in the lab experiments of Nielsen et al. 2016

Line 263-264: "For the purpose of illustrating the general relation between stress, dissipated fracture energy and rupture propagation," This purpose could also be reached by discussing results reported from the numerical simulations of Okubo et al. 2019.

Line 312-314: "Our results show that when dissipated fracture energy is not considered in the superficial portion of the fault, no significant rupture velocity reduction is observed." This is mainly the case because other important aspects of the shallow layer of sediments, such as the smaller seismic wave velocity, are not included in the model.

Line 380: "under the slip-weakening curve" → "between the slip-weakening curve and residual friction"

Figure 7a: It is hard to see the initial path towards the peak value. Maybe this plot can render better if displacements are displayed with a logarithmic scale. What is the origin of the hardening observed at the end of the blue curve? It would be interesting to also include the plots for the other experiments done at different confining pressure in Supplementary Information.

(Remarks on code availability)

The code is accessible and provides a clear README file. I could not run the scripts though, because I do not have a license to use Wolfram Mathematica.

Reviewer #3

(Remarks to the Author)

De Paola et al. focus on on- and off-fault energy partitioning during co-seismic slip. The authors combine field observations, experiments, and modeling to better constrain energy partitioning in soft sediments. Their analysis centers on a single fault plane, which they argue was activated by three events of $M \geq 6$, based on Allen (1982). They quantify off-fault damage and average slip for these events. To further constrain on-fault energy dissipation during co-seismic slip in soft sediments, they conducted a series of high-velocity rotary shear experiments. These experiments revealed significant anomalies between the measured fracture energy and the values expected from seismic studies (e.g., Cocco et al., 2023). By estimating fracture energy from the near-surface fault damage zone, they obtained results more consistent with overall seismic measurements, suggesting that most fracture energy in soft sediments is consumed by off-fault damage. Finally, they develop a linear elastic fracture mechanics model to describe co-seismic energy dissipation as a function of rupture velocity and fracture energy. Overall, the authors set an ambitious goal and present compelling arguments regarding energy dissipation in soft sediments. Their work is well-crafted and thoughtfully executed. However, to enhance its clarity and impact, it is essential to explicitly address error bars in their measurements and analysis, as the uncertainties could span orders of magnitude. Additionally, the authors could help readers by clearly outlining the rationale and practical approach of the paper earlier in the text. This would allow readers to fully appreciate the novelty of the approach. I recommend minor revisions.

Major comments:

The observation that a single seismic breccia layer represents an event $\geq M6$ is significant and warrants further elaboration. This is a crucial stage of the paper and affects the other following steps. I think citing reference 31 alone may not be good enough. Could the authors provide a more detailed explanation, based on ref. 31, to substantiate this claim?

The authors calculate the off fault fracture energy based on section A in the methods after measuring the co seismic off fault damage around fault LIF 1.1. Assuming the event dimensions are 1 m slip and a 10 km rupture diameter, how reliable is it to constrain off-fault damage from field measurements of a ~50 m zone?

The authors report 14 experiments but present data from only four samples in Figure 7. Could the authors clarify why fracture energy measurements were limited to the 9 MPa experiments (I believe it is equivalent to 500 m depth)? Additionally, what is the rationale for reporting only four experiments, and how were these selected? Given that fracture energy is normal-stress-dependent, it would be helpful to address the implications of varying normal stresses on the measured energy dissipation.

The authors should explicitly address the expected errors at each stage of their analysis. For example:

The uncertainty in identifying $M6$ seismic breccia layers (a magnitude 6.5 event may yield significantly different partitioning). Off-fault damage measurements during co-seismic periods.

Experimental error in laboratory measurements, including normal stress effects and experimental off-fault damage, even if

reported as minor.

The section on field measurements provides extensive detail about materials and settings before the reader has a clear understanding of how these relate to the research questions outlined in the introduction. One option for example is introducing Figure 8 earlier in the paper, alongside a schematic workflow, to guide readers through the combined approaches and data analysis.

The section on field measurements provides extensive detail about materials and settings before the reader has a clear understanding of how these relate to the research questions outlined in the introduction. For instance, introducing Figure 8 earlier in the paper, along with a schematic workflow, could help guide readers through the combined approaches and data analysis more effectively.

Minor Comments

Lines 71–73: Consider citing Alsop and Marco (2013) for readers unfamiliar with gravitational slumping.

Figure 3, Lines 631–634: Clarify what "LIF" stands for (likely Lisan Fault). Additionally, the reported maximum displacement of 337 cm is unclear; following the colored horizontal lines, the offset appears closer to 20 cm. Similarly, lines 207–211 require clarification on how the 1.85 m offset is observed in Figure 5.

Lines 214–216: Could the authors provide absolute ages for the faulting events? If the 2 ky period is derived from the Lisan Formation lamina, I guess the absolute age is also possible.

Figure 7: The weakening phase for brine-saturated samples is unclear in the figure. As these are likely critical for the energy balance, a better graphical presentation is needed.

Lines 219–220: Clarify the calculation of fracture energy. The weakening distance in the brine-saturated Lisan gouge from Figure 7 is reported as 6 cm, not 0.62 m. Additionally, the corresponding depth is stated as <1 km, but it appears closer to 500 m. Be more explicit here, as this part is confusing but critical for the reader's understanding.

Supplementary Table II: Include the equation used to calculate fracture energy in the caption for clarity.

Lines 180–181: Can the authors better explain the mechanism behind the faster weakening in brine-saturated gouges. What is the physical process?

Lines 170–171: Clarify (at least to me) why fracture energy analysis focused solely on fault LIF 1.1. Was it the only fault with seismic breccia layers?

Methods, Equation 3: If the authors are applying an intact failure criterion, state this explicitly. If not, explain why experimental data were not used.

Typos

Line 661: Use "Fig. b" instead of "fig. B."

Lines 686, 689, 694: Consistently format MPa.

Lines 111, 209: Use "localized."

SI Figure 8: Clarify the reference to "blue."

Line 646: Use "Fig. b" instead of "fig. b."

Ref:

Allen, J.R.L., 1982. *Sedimentary Structures: Their Character and Physical Basis*. Elsevier, New York.

Alsop, G. I., & Marco, S. (2013). Seismogenic slump folds formed by gravity-driven tectonics down a negligible subaqueous slope. *Tectonophysics*, 605, 48-69.

Cocco, M., Aretusini, S., Cornelio, C., Nielsen, S.B., Spagnuolo, E., Tinti, E., Di Toro, 480 G., Fracture energy and breakdown work during earthquakes. *Annu. Rev. Earth Planet. Sci.* 481 2023. 51:217–52 (2023).

(Remarks on code availability)

Version 1:

Reviewer comments:

Reviewer #2

(Remarks to the Author)

I thank the authors for the careful revision of their paper. The revision has significantly improved and clarified the paper by stating the goals and scope of the theoretical model and reshaping the discussion section, including framing the results of the paper in the context of past works.

I recommend publication of the manuscript. Below are minor points that the authors can decide to address or not:

I.19 (abstract): "that omitting fracture energy has little effect on shallow rupture velocity": The meaning of this sentence is not clear, particularly at the stage of the abstract. I recommend to clarify what is implied here. If I understand correctly, it refers to the fact that the low initial stress stored in the soft sediments is not sufficient to affect the rupture velocity, whereas also accounting for the higher fracture energy due to off-fault dissipation causes significant rupture deceleration.

I.266 and I.458: mode II corresponds to "in-plane shear" and not to "anti-plane" which is mode III.

I.526: "can be downloaded" is written twice.

Figure 9b: It can be interesting to also include in this plot four additional points that correspond to your laboratory friction experiments shown in Figure 7a.

(Remarks on code availability)

Reviewer #3

(Remarks to the Author)

In the revised manuscript, De Paola et al. have addressed most of my previous comments. The new version is clearer and significantly improved. As I mentioned earlier, this is a unique and important contribution that unifies multiple aspects of earthquake physics under one framework. The authors have done an excellent job integrating four major approaches in the field: field geology, seismology, laboratory experiments, and modeling.

One of the most impactful insights of this study is the finding that off-fault energy dissipation in soft sediments, such as those in the Lisan Formation, can account for up to 85% of the total fracture energy.

That said, I have one remaining concern regarding the treatment of uncertainty in the interpretation of fracture energy, particularly in light of my previous Comment #4 (R3) from the first review round. I recognize that my earlier remark may not have been clear enough, and I apologize for the lack of precision.

To be specific: in Figure 9b (referencing Cocco et al., 2023), the estimate of 1–2 MJ/m² for total breakdown work from a displacement of ~0.6 m carries a potential uncertainty of roughly an order of magnitude. This large variance needs to be acknowledged more clearly. I fully support the authors' choice to use an average value (if I understand correctly), whether derived from seismic data, laboratory experiments, or another method and as long as the decision is explicitly justified. I do not think this point justifies another round of revision. In my view, the paper is ready for publication. However, I believe that a short statement acknowledging the typical range of uncertainty in fracture energy estimates would improve the transparency and reliability of the work, especially for non-specialist readers.

I have three additional, minor suggestions that I believe would improve the clarity and accessibility of the paper:

Figure 7: The reported peak friction values of 0.30–0.36 are difficult to visually confirm. I suggest increasing the line width or using a more distinct color/contrast to make the friction peak more visible.

LEFM model explanation: In Section B of the Methods, I recommend adding a short sentence at the beginning to explain that, for dynamic rupture to occur, the energy flux G must equal the energy required to fracture the surface Γ . This would help readers who are less familiar with fracture mechanics understand the physical basis of the model.

Supplementary Table 1: Please consider adding the calculated energy values for the principal slip zone (PSZ) to Supplementary Table 1. Currently, these data are not directly accessible to the reader and would be useful for completeness and reproducibility.

Overall, I appreciate the authors' careful work on this revised version. I am satisfied with the manuscript as it now stands and believe that these final clarifications will further strengthen its impact.

(Remarks on code availability)

I used the code for reproducing Fig. 10. I added an additional comment to the authors to add the energy calculations in Table 1 because the raw experimental data is not available.

Answer to Reviewers' comments on De Paola et al. paper submitted to Nature Communications

We have carefully addressed the comments provided by the reviewers and are grateful for their constructive feedback and positive remarks.

We believe that responding to the reviewers' main criticisms has helped us to significantly strengthen and clarify the ideas presented in the paper.

Below, we provide a point-by-point response to each of the reviewers' comments (reproduced in italics). As requested, all changes and additions to the main text are highlighted in red in the revised version of the manuscript.

Revised manuscript length check:

Abstract: 150 words;

Main text: 4393 words;

Methods: 1587 words;

Figures: 10 Figures;

Number of references cited: 55.

ADDRESSING OF REVIEWERS' COMMENTS

Reviewer #1 (Remarks to the Author):

Reviewer 1 (R1): This is a really great paper, well written, with conclusions well supported by the data and modeling. I love the combination of field measurements, lab experiments and modeling to address the contribution to the energy budget from off-fault damage.

I found I did not need to edit the paper at all, just noting in 2 spots where a reference might be added.

Authors (A): We thank the reviewer for recognizing the strength of the multidisciplinary approach adopted in our work. As suggested by the reviewer, we have added the referenced citation (already included in our reference list) in the two locations in the annotated PDF (lines 221 and 501 in the revised manuscript), and also in the Discussion at lines 341 and 354, when we discuss our findings in the context of previous, relevant work.

R1: For a long time it seemed like workers in this field assumed fracture energy was effectively negligible with most/all non-radiated energy being converted to heat. In fact, there is very little, if any, evidence in nature for this assumption about heat. In more recent times, evidence is growing that fracture energy is quite significant when including off-fault damage. My research group is tackling this topic in systems exhumed from brittle-ductile transition depths, and it is really nice to see this study of the near-surface problem. I love the field photos - clearly an excellent place to do this kind of work. I am not surprised to see the large off-fault energy sink in these soft sediments, and I think the numerical simulations predicting that such large off-fault energy dissipation can slow down and arrest ruptures in soft sediments are sound.

I recommend publication in Nature Communications with minimal revision. This is an important topic, and this paper makes an excellent contribution.

A: We thank the reviewer for their supportive comments on the approach adopted, as well as the results and arguments presented in the manuscript.

Reviewer #2 (Remarks to the Author):

Reviewer 2 (R2): *The authors study the seismic behaviour of a layer of soft and poorly lithified sediments deposited near the Earth's surface. This scientific question is important to better understand and assess the risk of surface-rupturing earthquakes, which are still poorly understood. To do so, the authors studied an exhumed fault zone along the Dead Sea transform, referred to as the Masada Fault Zone (MFZ). The MFZ is particularly interesting as it cuts through a layer of poorly consolidated sediments deposited at the bottom of the Lake Lisan (the precursor of the current Dead Sea). The authors conducted field work to study the structure of fault zone in this sedimentary Lisan Formation (LF). An important observation reported by this study is that even the material within the principal slip zone (PSZ) shows little grain abrasion and fracturing, which suggests that the seismic deformation mostly arises through grain-scale particulate flow. To test this observation, the authors performed frictional experiments at seismic velocity conditions on sediments of the LF. They showed that the saturation of the sediments with brine is essential to reproduce the deformation and microstructure observed in the field. An important consequence on this particulate flow mechanism is the associated low value of residual friction and fracture energy measured during the experiments. To estimate the total energy dissipated during earthquake rupture, the authors also performed a survey of the damage observed in the field along a transect surrounding the PSZ, from which they estimated the amount of off-fault energy dissipated during the three major seismic events that arose along the MFZ. The comparison with the energy dissipated inside the PSZ (measured in the lab experiments) reveals that co-seismic fracture energy in these soft sediments is mainly due to off-fault dissipation (up to 85%). The paper concludes by presenting results from a model of shear fracture used to demonstrate the effect of larger off-fault fracture energy on the crack speed.*

The research question investigated in the paper is important and the authors conducted convincing field and laboratory works to present original results on the seismic behaviour of soft and poorly consolidated sediments deposited near the surface of the MFZ. The deformation mechanism within the PSZ and the very large portion of off-fault damage are all important observations, different from what is expected for fault zone across more competent rock. In my opinion, the present study could be an important contribution to the field.

Authors (A): We thank the reviewer for their insightful summary of our results, which captures the key implications of our findings, and for recognizing their potential contribution to the field. We also appreciate the reviewer's constructive criticisms, which helped us strengthen the manuscript's structure and clarify its core ideas.

R2: *However, the discussion section of the paper needs some major transformation. At the moment, it is mostly dedicated to present results from a fracture mechanics model rather than summarizing these important observations made from the field and lab works, and discussing them in the context of previously published works. The proposed model is not well-suited to study the problem of shallow layer of soft sediments and takes the place of important discussion of the original contribution of the paper. For these reasons, I recommend a major revision of the manuscript, particularly its discussion section.*

A: We appreciate the reviewer's constructive feedback and have substantially revised the Discussion section in response. Specifically, the presentation of the model and its results has been relocated from the Discussion to the Results section, under the new subheading "Energy dissipation in shallow rupture: The role of off-fault damage". In this revised section, we clarify both the limitations and the primary objectives of the modelling approach. The Discussion has been refocused to highlight the most relevant and original contributions of our findings, in line with the reviewer's suggestions. A detailed point-by-point response to the reviewer's comments is provided below.

R2 – Major points:

R2: *In the discussion, the authors present results based on the model of Fossum and Freund 1973. This fracture mechanics model is derived using boundary conditions and assumptions that are in contradiction with the set-up of interest shown in Figure 10.*

A: In our responses below, we outline the changes made to the structure and content of the manuscript to address the reviewer's three main concerns regarding specific boundary conditions and assumptions in the model.

R2: *First the model assumes an unbounded solid and cannot capture the effect of the free surface, which is important in the dynamics of surface-rupturing faults.*

A: The reviewer is correct in noting that the analytical model does not account for free surface effects, but instead captures only the influence of changes in dissipated energy due to cumulative on- and off-fault damage. We have clarified this limitation in the revised Results section (lines 266–277), where the model and its outcomes are now presented. This point is further emphasized in the revised Discussion section (lines 345–349).

R2: *Second, it assumes homogeneous bulk properties and cannot capture the strong contrast of material properties that exists between the layer of sediments and the bedrock. It seems, for instance, highly unlikely that the rupture will propagate across the sediments at speed close to the Rayleigh wave speed of the bedrock as shown in Figure 10.*

A: The reviewer is correct that our current model setup does not incorporate material heterogeneity between the shallow and deeper sections of the fault. Previous studies have

indeed attributed the slow rupture velocities observed in the shallow portions of megathrusts – such as in tsunami earthquakes (e.g., Bilek et al., *Nature*, 1999) – to differences in rigidity between soft shallow sediments and deeper bedrock.

However, as shown in Equation 18 (Methods B – Modelling of Rupture Velocity), rupture velocity is also strongly influenced by the dissipated fracture energy. This consideration motivated our decision to isolate and examine the role of off-fault damage – a factor that has received relatively little attention in seismological modeling of shallow ruptures but that our results indicate plays a critical role in controlling rupture velocity. We have clarified this rationale in the revised Discussion (lines 349–354) and in Methods B – Modelling of Rupture Velocity (lines 458–469). In the revised text, we also explicitly acknowledge that other mechanisms influencing rupture propagation, such as fault zone heterogeneity, have been extensively explored in prior work.

R2: *Third, the model assumes linear elastic fracture mechanics conditions with questionable applicability to a material “which only allows support of very limited elastic stress before failure or ductile deformation” (quoting here lines 270-271 of the paper).*

A: We have clarified this point in the revised text (Methods B – Modelling of Rupture Velocity, lines 458–462). Our field observations reveal localized brittle failure along discrete fault branches within the Lisan Formation. Supporting this, laboratory triaxial tests conducted under brine-saturated, undrained conditions (Reference 52: Frydman et al., *Engineering Geology*, 2008) demonstrate that the Lisan material exhibits a brittle failure envelope at very low stress levels.

Based on these findings, we consider it appropriate – as a first-order approximation – to treat the Lisan units as rigid-elastic bodies prior to brittle failure. Accordingly, we apply Linear Elastic Fracture Mechanics (LEFM), assuming linear elasticity, while explicitly accounting for the low stress regime and the reduced failure threshold imposed by limited confining pressure and cohesion.

R2: *I am not against the simplicity of the model, but the conclusions drawn from this model should be taken with much more care. In this context, presenting it as a “quantitative framework for shallow ruptures propagating through soft sediments” is a clear overstatement that should be nuanced.*

A: We thank the reviewer for their thoughtful critique. In response, we have revised the text to remove overstatements and have nuanced the *Discussion* section accordingly. While our model integrates field observations, laboratory data, and mathematical analysis, it is not intended to capture the full complexity of natural earthquake rupture processes. We have now clarified the scope and limitations of our simplified approach in the revised *Results* section (lines 266–277). In the *Discussion*, we also acknowledge that more comprehensive studies – incorporating complex fault geometries, material heterogeneity (e.g., reference 14: Okubo et al., 2019), and the effects of the free surface (e.g., references 47: Nielsen, 1998; and 48: Kozdon & Dunham, 2013) – have addressed these aspects in greater depth.

R2: Another issue with the model is that it is introduced in the discussion section, which is not the appropriate place to present model and results. Consequently, the paper missed a proper discussion of the original and important results from lab and field works in the context of previous works. How do the measured on-fault and off-fault dissipations compare with other field/lab measurements made on more competent fault rock? How does the portion of off-fault dissipation (85%) compare with values typically assumed/observed in the literature? These are few examples of important aspects that must be discussed in this section of the paper. In my opinion, the fracture mechanics model can simply be removed from the main text. If its purpose is to illustrate the relation between stress, off-fault dissipation and rupture propagation, it should rather be substituted by a discussion of published modelling results, such as the one of Okubo et al. 2019 that quantified how off-fault damage slows down rupture and reduces the radiation efficiency.

A: We agree that introducing the model and presenting its results within the *Discussion* was not appropriate. However, we believe the modelling component provides valuable insights into the case study. In response to the reviewer's suggestions, we have retained this material but relocated the model description – including its purpose and limitations – and the presentation of results to a dedicated subsection in the *Results*: “Energy dissipation in shallow rupture: The role of off-fault damage” (lines 265–308).

As advised, we have substantially restructured the entire *Discussion* to focus on the implications of our novel field observations and energy budget estimates. These results are now discussed in comparison with previous estimates for more competent lithologies, allowing for a contextualised analysis.

In the second part of the *Discussion* (Lines 333 – 369), we have revised the text to temper overstatements regarding our modelling, clarify its scope, and highlight its broader relevance, while referencing more comprehensive models in the literature. We thank the reviewer for their insights, which have helped improve the focus and clarity of the *Discussion*, and better emphasize the main contributions of our study within the broader research context.

R2 – Minor points:

R2: As *Nature Communications* is a multidisciplinary journal, more effort could be made to clarify some concepts and terminology in the paper (see also the line-by-line comments hereafter). For example, the sketch of Figure 10a could be already included in Figure 1 to provide a generic illustration of a surface-rupturing fault crossing a layer of sediments. The energy budget of earthquakes is another central aspect of the article that deserves further precision. For instance, I would recommend clarifying the physical meaning of these different energy contributions that are listed at lines 200-201 (for example: “(...) into the fracture energy E_G that describes the dissipated energy required to bring fault friction down to its residual value, the energy E_H dissipated by slip at residual friction mainly in the form of heat and the energy radiated by seismic waves E_R ”).

A: Following the reviewer's suggestion, we have revised lines 214–217 to clarify the physical meaning of the earthquake energy budget components. These modifications aim to ensure that the concepts are accessible to *Nature Communications* readers with broad scientific backgrounds.

Regarding the suggestion to incorporate Figure 10a into Figure 1, we respectfully prefer to keep the figures separate. Figure 1 illustrates the actual geological setting of the fault zones in the study area, while Figure 10a serves as a conceptual representation of the boundary conditions used in our modelling. Merging them could compromise the clarity and distinct purposes of each figure.

R2 – Line by line comments and suggestions:

R2 – Abstract line 18-20: *“Our numerical simulations predict that such high amount of off-fault energy dissipation in the shallow portion of the fault can slow down and arrest ruptures in soft sediments.” This is an example of claim based on the fracture mechanics model that should be toned down or removed. This is also the only place of the paper that presents this model as “numerical simulations”.*

A: As requested by the reviewer, we have revised and toned down the text in the abstract (lines 18–20) to more accurately reflect that the modelling is specific to our case study and limited to the effects of off-fault damage on rupture velocity in faults hosted in soft sediments.

R2 – Line 29: *“The total potential energy of an earthquake” → “The potential energy released during an earthquake”*

A: Text changed as suggested by the reviewer.

R2 – Line 45-46: *“(e.g., energy sinks related to on-fault vs. off-fault deformation)” It is not clear, especially for a broad audience, what “energy sinks” is referring to in this sentence.*

A: The text at lines 45–48 has been revised to improve accessibility for a general audience, while retaining essential subject-specific terminology to preserve scientific accuracy.

R2 – Line 109-111: *Is the energy dissipated along subsidiary slip surfaces accounted for in the in-fault or in the off-fault fracture energy?*

A: The energy dissipated by subsidiary slip surfaces is included in the on-fault fracture energy estimates, as these features connect to or branch from the principal slip zones. Hence, we included them in the description of the fault core structures. In contrast, other shear structures that contribute to off-fault fracture energy are isolated within the damage zone and are not connected to the main slipping zone in the fault core. These are discussed separately in the *Off-fault deformation* section.

R2 – Line 119-121: *“It is particularly striking that even within the PSZ, some of the aragonite crystals retain their delicate radial, rosette-like arrangements of needles during deformation”*

Very nice indeed! Could this indicate that the particulate flow has a very localized flow profile, where most of the flow experiences little shear?

A: Our preferred interpretation of the data presented in Figure 4 is that particulate flow is localized within the principal slip zone (PSZ), where intense shear is indicated by the slip zone parallel alignment of elongated aragonite acicular crystals and the “cortex” rosette structures, which likely formed through grain rolling during shear. While we cannot entirely rule out the possibility of en-masse flow being localized in the PSZ, we believe that the microstructural evidence more strongly supports shear-driven deformation within this zone.

R2 – Line 170-171: *“These experiments allow us to assess how sensitive co-seismic fault strength is to burial depth in the uppermost part of the crust.” From your experiments it seems that brine saturation has a much stronger impact than burial depth. This is an interesting aspect that is worth highlighting and developing further in the discussion section.*

A: We have addressed this point in the revised *Discussion* section (lines 318–325).

R2 – Line 203-206: *“The total mechanical, or fracture, energy (E_G) dissipated by fault processes during rupture propagation is equal to the sum of the mechanical work done on-fault, within the PSZs (E_G^{PSZ}) of the fault core, and off-fault ($E_G^{Off-fault}$), in the surrounding damage zone.” This is unclear, because the work of residual friction associated to heat dissipation also corresponds to “mechanical work done on-fault”. “Breakdown work” as discussed in Cocco et al. 2023 might be a more precise way to describe and refer to this dissipation.*

A: The reviewer is correct, at line 220 and elsewhere in the main text, we have replaced “mechanical work” with the more correct “breakdown work”.

R2 – Line 222: *“This result agrees with the minimal grain-scale fracturing observed” It could be insightful to also discuss and compare these measurements of E_G to the fracture energy measured from the experiments at room-humidity that show evidence of fracturing and grain size reduction.*

A: While we have estimated fracture energy from the mechanical data, a more thorough assessment from the room-humidity experiments would require benchmarking against grain size analyses from both thin sections and sheared powders. This is part of ongoing work that we plan to present in a longer-format paper to be submitted in the coming months.

R2 – Line 228: *“very small, almost negligible when compared to” → “about one order of magnitude smaller than”*

A: Text changed at line 245 as suggested by the reviewer.

R2 – Line 230: *It would be insightful to compare and discuss the on-fault fracture energy measured in this study with the one reported from the references listed at line 230 in the text, particularly the one measured in the lab experiments of Nielsen et al. 2016.*

A: We refer to Cocco et al. (2023), which provides a more comprehensive and appropriate comparison than citing Nielsen et al. (2016) alone, as it incorporates the results from Nielsen (2016) along with more recent estimates. As noted in line 753, we also comment that the energy estimates from our study are compatible with the seismological estimates reported by Cocco et al.

R2 – Line 263-264: *“For the purpose of illustrating the general relation between stress, dissipated fracture energy and rupture propagation,” This purpose could also be reached by discussing results reported from the numerical simulations of Okubo et al. 2019.*

A: This is a valid point, and we believe we have addressed it by revising the *Discussion* section in line with the reviewer’s suggestions. In our previous responses, we also explained our rationale for retaining the simplified modelling component in the manuscript. Additionally, at lines 341–344 and 355–364 of the *Discussion*, we place our findings on rupture propagation in shallow faults in the context of previous studies, including that of Okubo et al. (2019), which is more relevant to seismic rupture in deeper crustal settings.

R2 – Line 312-314: *“Our results show that when dissipated fracture energy is not considered in the superficial portion of the fault, no significant rupture velocity reduction is observed.” This is mainly the case because other important aspects of the shallow layer of sediments, such as the smaller seismic wave velocity, are not included in the model.*

A: We agree that the original text lacked clarity in explaining the main objective of our analysis. As outlined in our earlier responses, we intentionally simplified the model by neglecting other controls on rupture velocity to isolate and emphasize the role of dissipated fracture energy, particularly that associated with fault structure – such as off-fault energy dissipation due to fracturing in the damage zone.

Although dissipated fracture energy is theoretically recognized as a key factor in controlling rupture velocity (see Equation 18 in *Methods B*), it has been largely overlooked in previous seismological studies. Our results underscore that this energy component is not negligible in shallow faults, and therefore should not be excluded from realistic seismological modelling implying that rupture velocity only scales with acoustic wave velocity of soft sediments.

A detailed response to this point is provided above, and the revised text in the *Discussion* (lines 347–354) now clarifies this rationale more explicitly.

R2 – Line 380: *“under the slip-weakening curve” → “between the slip-weakening curve and residual friction”*

A: Text changed at line 408, as suggested by the reviewer.

R2 – Figure 7a: *It is hard to see the initial path towards the peak value. Maybe this plot can render better if displacements are displayed with a logarithmic scale. What is the origin of the hardening observed at the end of the blue curve? It would be interesting to also include the plots for the other experiments done at different confining pressure in Supplementary Information.*

A: To maintain clarity and focus for a general audience, we prefer to keep the main experimental graphs simple and not include detailed shear strength evolution with slip. However, as the reviewer notes, this information may be of interest to readers with more expertise in fault mechanics. Key mechanical parameters from all experiments are summarized in Supplementary Table 1. In response to the reviewer's request, we have also included the full shear strength vs. slip curves in the Supplementary Information as Supplementary Figure 9. Supplementary Figure 9E-F now also includes the initial paths of friction evolution to peak values of the experimental results shown in Figure 7 of the main text.

A small amount of restrengthening is always observed in high velocity friction experiments during the deceleration phase to arrest. Regarding the more pronounced restrengthening observed in experiment 525 (Fig. 7a), this results from gouge loss during the final 25 cm of slip, as material was trapped between the confining Teflon ring and the metal sample holder. Importantly, this did not affect the experiment itself, as no significant gouge thickness variations were observed up to 1.45 m of slip.

Reviewer #2 (Remarks on code availability):

R2: *The code is accessible and provides a clear README file. I could not run the scripts though, because I do not have a license to use Wolfram Mathematica.*

A: The code provided in the repository is available in both Mathematica's native format and as a readable sequence of commands in a PDF file. To reproduce the results without a Mathematica license, the commands can be copied and run via the freely accessible online platform <https://www.wolframalpha.com/>. Additionally, for readers interested in conducting more extensive analyses or modifying the code, it would be possible to translate the Mathematica commands into other programming languages. The language is intuitive, and detailed information on the syntax and functions is readily accessible through the WolframAlpha search tool.

Reviewer #3 (Remarks to the Author):

Reviewer 3 (R3): *De Paola et al. focus on on- and off-fault energy partitioning during co-seismic slip. The authors combine field observations, experiments, and modeling to better constrain energy partitioning in soft sediments. Their analysis centers on a single fault plane, which they argue was activated by three events of $M \geq 6M$, based on Allen (1982). They quantify off-fault damage and average slip for these events. To further constrain on-fault energy dissipation during co-seismic slip in soft sediments, they conducted a series of high-velocity rotary shear experiments. These experiments revealed significant anomalies between the measured fracture energy and the values expected from seismic studies (e.g., Cocco et al., 2023). By estimating fracture energy from the near-surface fault damage zone, they obtained results more consistent with overall seismic measurements, suggesting that most fracture*

energy in soft sediments is consumed by off-fault damage. Finally, they develop a linear elastic fracture mechanics model to describe co-seismic energy dissipation as a function of rupture velocity and fracture energy.

Overall, the authors set an ambitious goal and present compelling arguments regarding energy dissipation in soft sediments. Their work is well-crafted and thoughtfully executed. However, to enhance its clarity and impact, it is essential to explicitly address error bars in their measurements and analysis, as the uncertainties could span orders of magnitude. Additionally, the authors could help readers by clearly outlining the rationale and practical approach of the paper earlier in the text. This would allow readers to fully appreciate the novelty of the approach.

I recommend minor revisions.

Author (A): We thank the reviewer for their thoughtful summary, which highlights the key implications of our findings. We also appreciate the constructive feedback, which has helped us improve the clarity and strength of the manuscript. Below, we respond in detail to each of the reviewer's comments.

R3 – Major comments:

R3: *The observation that a single seismic breccia layer represents an event $\geq M6$ is significant and warrants further elaboration. This is a crucial stage of the paper and affects the other following steps. I think citing reference 31 alone may not be good enough. Could the authors provide a more detailed explanation, based on ref. 31, to substantiate this claim?*

A: We apologize to the reviewer for any confusion caused by a lack of clarity in the original text, which may have led to the misunderstanding that each individual seismic breccia layer observed along the Masada Fault Zone corresponds to a single seismic event with magnitude $M \geq 6$, solely based on the theoretical work of Allen (1982; cited as reference 31 in the manuscript).

In fact, our assumption that the three distinct seismic breccia layers observed along the synsedimentary coseismic fault LIF 1.1 reflect three separate rupture events of at least magnitude $M = 6$ is based primarily on the detailed paleoseismological study of the Masada Fault Zone by Marco and Agnon (2005; reference 18). These authors applied high-resolution, sub-centimeter-scale stratigraphy along with conventional paleoseismic techniques to natural exposures of Late Pleistocene lacustrine laminates near Masada. Building on earlier work (Marco and Agnon, 1995; reference 17), they examined columnar sections on both sides of several well-exposed, N-striking normal faults – including LIF 1.1, the fault analyzed in our study. Their results revealed three distinct seismic slip events on these faults, each with approximately 60 cm of co-seismic slip, the most recent occurring about 25 kyr ago.

The inference that these events were at least magnitude $M = 6$ is grounded in empirical relationships between fault slip and event magnitude, such as those proposed by Sibson (Journal of Structural Geology, 1989), which relate ~ 0.4 m of slip to $M = 6$ events. Additional

support comes from seismological studies (Bonilla et al., 1984; Wells & Coppersmith, 1994) showing that surface rupture is generally limited to earthquakes of magnitude ≥ 5.5 –6. Since the breccia layers are directly associated with surface rupture feature (see references 18 and 29), the inferred magnitudes are consistent with both the geological and empirical evidence. Finally, the theoretical work by Allen (1982; reference 31) supports the conclusion that events of $M \geq 6$ have sufficient energy to generate seismites and other structures observed in the Masada Fault Zone.

To address the reviewer's concern, we have revised the text between lines 99 – 109 and 222–233 to clarify this interpretation.

R3: *The authors calculate the off fault fracture energy based on section A in the methods after measuring the co seismic off fault damage around fault LIF 1.1. Assuming the event dimensions are 1 m slip and a 10 km rupture diameter, how reliable is it to constrain off-fault damage from field measurements of a ~50 m zone?*

A: If we understand the reviewer's question correctly, they are asking how reliable it is to estimate off-fault fracture energy across a fault section (e.g., a tens-of-meters-wide damage zone) for a fault that extends over kilometers in length along strike.

While the reviewer raises a valid point, we base our observations on the exposures of the Masada Fault Zone (MFZ) that allow for mapping of damage zone thickness along strike in various exposures, which are easily accessible along incised canyons. We note that earthquake surface ruptures like those along the MFZ are rarely preserved due to rapid degradation from surface weathering or inaccessibility (e.g., being located below sea level in subduction zones). For this reason, observations from the MFZ are almost unique and extremely valuable for shedding light on the relative contribution of off-fault energy dissipation to the energy balance of near-surface ruptures in soft sediments.

R3: *The authors report 14 experiments but present data from only four samples in Figure 7. Could the authors clarify why fracture energy measurements were limited to the 9 MPa experiments (I believe it is equivalent to 500 m depth)? Additionally, what is the rationale for reporting only four experiments, and how were these selected? Given that fracture energy is normal-stress-dependent, it would be helpful to address the implications of varying normal stresses on the measured energy dissipation.*

A: In response to similar comments from Reviewer 2 (see above), we have now included graphs for all experiments in the Supplementary Information (see Supplementary Figure 9). The main mechanical parameters were already summarized in Supplementary Table I.

Fracture energy estimated from high-velocity friction experiments depends on the slip-weakening distance D_w (see conceptual Figure 8b in the main text). For the brine-saturated gouges studied, D_w is comparable for experiments conducted at normal loads of 9 and 18 MPa (see Supplementary Figure 9 and Table I). This indicates that, within the near-surface conditions investigated, normal load (9 vs. 18 MPa) has no significant effect on fracture energy.

We chose to present a smaller set of representative experiments in Figure 7 to enhance readability and to help readers without specialized expertise in rock mechanics grasp the key findings. This selection is supported by the strong reproducibility observed across multiple runs, now fully shown in Supplementary Figure 9.

R3: *The authors should explicitly address the expected errors at each stage of their analysis. For example:*

- a) *The uncertainty in identifying M6 seismic breccia layers (a magnitude 6.5 event may yield significantly different partitioning).*
- b) *Off-fault damage measurements during co-seismic periods.*
- c) *Experimental error in laboratory measurements, including normal stress effects and experimental off-fault damage, even if reported as minor.*

A: a) In our reply above we have addressed how seismites have been used to distinguish seismic events occurring on a single fault, primarily based on the detailed paleoseismological study of the Masada Fault Zone by Marco and Agnon (2005; reference 18). We emphasize that the estimation of on- and off-fault fracture energy does not depend on any magnitude estimate or assumptions, but rather on mechanical data (on-fault component) and field measurements (off-fault component) collected.

b) We have now clarified in the caption of Supplementary Table II that the measurement error for damage zone fractures in the field is less than 1 mm.

c) As explained above, within the range of normal stress investigated in this study, there is no observed dependence of slip-weakening distance D_w — and thus fracture energy — on normal stress. The error in friction measurements depends on the noise level in each experiment and generally ranges between 0.01 and 0.03. This is a minimal error that does not significantly affect the estimation of on-fault fracture energy. Such error levels are typical in high-velocity friction experiments. We have now clarified this in the figure caption of Supplementary Figure 9, where the mechanical dataset is presented.

R3: *The section on field measurements provides extensive detail about materials and settings before the reader has a clear understanding of how these relate to the research questions outlined in the introduction. One option for example is introducing Figure 8 earlier in the paper, alongside a schematic workflow, to guide readers through the combined approaches and data analysis.*

A: It would be difficult to introduce Figure 8 earlier in the main text, as its full significance may not be clear to readers — particularly those without specialized expertise — without first presenting the field data on fault structure, the experimental results showing the evolution of friction, and the fundamental concepts of the earthquake energy budget. However, we acknowledge the reviewer's valuable point and have addressed it by adding explanatory text towards the end of the Introduction (lines 50–56), outlining the workflow of the manuscript.

R3 – Minor Comments

R3 – Lines 71–73: Consider citing Alsop and Marco (2013) for readers unfamiliar with gravitational slumping.

A: Reference added as suggested by the reviewer.

R3 – Figure 3, Lines 631–634: Clarify what "LIF" stands for (likely Lisan Fault). Additionally, the reported maximum displacement of 337 cm is unclear; following the colored horizontal lines, the offset appears closer to 20 cm. Similarly, lines 207–211 require clarification on how the 1.85 m offset is observed in Figure 5.

A: We have corrected the text to "LIF2.4" at line 698, which is the code name of the fault. The studied faults in the Lisan Formation are synsedimentary, and their displacement varies along dip. The deeper portions of the faults record greater displacement than the upper portions, as they have experienced more slip events. This is supported by the along-dip displacement profiles shown in the Supplementary Information (see Supplementary Figure 5b for LIF2.4). Figure 3 presents only a partial view – specifically, the upper part – of fault LIF2.4, where displacements range from approximately 20 cm (yellow marker) to 185 cm (green marker). In contrast, Figure 6b shows the full extent of the fault, with a maximum displacement of 337 cm (yellow marker in the lower portion), consistent with the detailed along-slip profile in Supplementary Figure 5b. The same considerations apply to the 1.85 m maximum displacement observed along fault LIF1.1, as previously detailed in Marco and Agnon (2005; reference 18).

R3 – Lines 214–216: Could the authors provide absolute ages for the faulting events? If the 2 ky period is derived from the Lisan Formation lamina, I guess the absolute age is also possible.

A: We have addressed this issue in our earlier reply concerning the estimation of co-seismic slip per single event (see revised text at lines 222–233), by referring to the detailed paleoseismic study of Marco and Agnon (2005; reference 18).

R3 – Figure 7: The weakening phase for brine-saturated samples is unclear in the figure. As these are likely critical for the energy balance, a better graphical presentation is needed.

A: This issue is now addressed in Supplementary Figure 9, where the full dataset is presented. For further details, please refer to our response to Reviewer 2, who raised a similar comment. We note here that Supplementary Figure 9E-F now also includes the initial paths of friction evolution to peak values of the experimental results shown in Figure 7 of the main text.

R3 – Lines 219–220: Clarify the calculation of fracture energy. The weakening distance in the brine-saturated Lisan gouge from Figure 7 is reported as 6 cm, not 0.62 m. Additionally, the corresponding depth is stated as <1 km, but it appears closer to 500 m. Be more explicit here, as this part is confusing but critical for the reader's understanding.

A: We believe it is accurate to state in the main text that we used the mechanical data to calculate the on-fault fracture energy for an event with 0.62 m of slip. The fact that slip amounts exceeding the slip-weakening distance (6 cm) do not contribute to fracture energy is implicit in the definition of fracture energy, as discussed in the text and illustrated in Figure 8.

A normal stress of 9 MPa acts on a fault dipping at approximately 60 degrees. This results in a lithostatic load of about 18 MPa, which – given the high porosity and low density of the Lisan formation – corresponds to a depth between 750 m and 1 km. Accordingly, we refer in the text to depths of < 1 km.

R3 – Supplementary Table II: *Include the equation used to calculate fracture energy in the caption for clarity.*

A: Thanks for the suggestion. In the caption to Supplementary Table II, we now refer to the Methods A section in the main text and to Supplementary Figure 10 for details about the calculations.

R3 – Lines 180–181: *Can the authors better explain the mechanism behind the faster weakening in brine-saturated gouges. What is the physical process?*

A: The mechanisms responsible for rapid weakening in brine-saturated gouges from the onset of slip have been thoroughly investigated in previous studies by our group (Bullock et al. 2015, cited as reference 45) and others (Faulkner et al. 2011 and Ujie et al. 2013, cited as references 43 and 44, respectively).

As the reviewer correctly notes, this topic may be of particular interest to readers with a strong background in fault mechanics. In response, we have revised the text at lines 193–195 to reference these published studies, which explore in greater detail the evolution of shear strength in saturated, clay-rich gouges. These works highlight the characteristic near-instantaneous weakening that occurs at the onset of slip.

R3 – Lines 170–171: *Clarify (at least to me) why fracture energy analysis focused solely on fault LIF 1.1. Was it the only fault with seismic breccia layers?*

A: Fault LIF1.1 is not the only fault studied that is associated with breccia layers. Therefore, most of the faults investigated are suitable for constraining and interpreting the structure and deformation mechanisms of near-surface co-seismic faults. However, for the fracture energy analysis, high-resolution paleoseismic data are required to precisely correlate individual seismite layers with specific seismic slip events on a given fault. These detailed correlations were beyond the initial scope of our project. Nonetheless, such high-resolution data were available for fault LIF1.1, based on the work of Marco and Agnon (2005; Reference 18). For this reason, we focused our fracture energy estimates on fault LIF1.1.

R3 – Methods, Equation 3: *If the authors are applying an intact failure criterion, state this explicitly. If not, explain why experimental data were not used.*

A: As noted in lines 393–399, we apply the Coulomb failure criterion using parameters derived by Frydman et al. (2008) based on triaxial loading experiments. In their study, cohesion was found to be very small – almost negligible – in its contribution to the estimated peak stress.

R3 – Typos

R3 – Line 661: Use "Fig. b" instead of "fig. B."

A: Corrected as suggested by the Reviewer.

R3 – Lines 686, 689, 694: Consistently format MPa.

A: Corrected as suggested by the Reviewer.

R3 – Lines 111, 209: Use "localized."

A: Corrected as suggested by the Reviewer.

R3 – SI Figure 8: Clarify the reference to "blue."

A: We are sorry but we are not sure about what to modify in SI Figure 8.

R3 – Line 646: Use "Fig. b" instead of "fig. b."

A: Corrected as suggested by the Reviewer.

Answer to Reviewers' comments on De Paola et al. paper submitted to Nature Communications

Dear Editor, we have carefully addressed the latest comments provided by the reviewers on our resubmitted manuscript.

Below, we provide a point-by-point response to each of the reviewers' comments (reproduced in italics). All changes and additions to the main text are highlighted in green in the revised final version of the manuscript.

ADDRESSING OF REVIEWERS' COMMENTS

Reviewer #2 (Remarks to the Author):

Reviewer 2 (R2): *I thank the authors for the careful revision of their paper. The revision has significantly improved and clarified the paper by stating the goals and scope of the theoretical model and reshaping the discussion section, including framing the results of the paper in the context of past works.*

I recommend publication of the manuscript. Below are minor points that the authors can decide to address or not:

R2: I.19 (abstract): *"that omitting fracture energy has little effect on shallow rupture velocity": The meaning of this sentence is not clear, particularly at the stage of the abstract. I recommend to clarify what is implied here. If I understand correctly, it refers to the fact that the low initial stress stored in the soft sediments is not sufficient to affect the rupture velocity, whereas also accounting for the higher fracture energy due to off-fault dissipation causes significant rupture deceleration.*

Authors (A): Following the reviewer's suggestion, we have revised the sentence at line 18 of the abstract to eliminate ambiguity and improve clarity.

R2: I.266 and I.458: *mode II corresponds to "in-plane shear" and not to "anti-plane" which is mode III.*

A: We thank the reviewer for spotting this. This is now corrected as pointed out by the reviewer.

R2: I.526: *"can be downloaded" is written twice.*

A: Typo spotted by the reviewer is now fixed.

R2: Figure 9b: *It can be interesting to also include in this plot four additional points that correspond to your laboratory friction experiments shown in Figure 7a.*

A: We appreciate the reviewer's suggestion; however, we prefer to keep Figure 9b in its current format. The experimental results shown in Figure 7a are already incorporated in the graph as part of the combined contribution of geological and experimental estimates (e.g., the red dot in the graph). Adding additional points may risk confusing the reader. The relative

contributions of the experimental and geological estimates are instead presented in Figure 9a, which explains the energy partition between on-fault (experimental) and off-fault (field) estimates.

Reviewer #3 (Remarks to the Author):

Reviewer 3 (R3): *In the revised manuscript, De Paola et al. have addressed most of my previous comments. The new version is clearer and significantly improved. As I mentioned earlier, this is a unique and important contribution that unifies multiple aspects of earthquake physics under one framework. The authors have done an excellent job integrating four major approaches in the field: field geology, seismology, laboratory experiments, and modeling. One of the most impactful insights of this study is the finding that off-fault energy dissipation in soft sediments, such as those in the Lisan Formation, can account for up to 85% of the total fracture energy.*

Author (A): We thank the reviewer for their appreciation of the revised version of the manuscript.

R3: *That said, I have one remaining concern regarding the treatment of uncertainty in the interpretation of fracture energy, particularly in light of my previous Comment #4 (R3) from the first review round. I recognize that my earlier remark may not have been clear enough, and I apologize for the lack of precision.*

To be specific: in Figure 9b (referencing Cocco et al., 2023), the estimate of 1–2 MJ/m² for total breakdown work from a displacement of ~0.6 m carries a potential uncertainty of roughly an order of magnitude. This large variance needs to be acknowledged more clearly. I fully support the authors' choice to use an average value (if I understand correctly), whether derived from seismic data, laboratory experiments, or another method and as long as the decision is explicitly justified.

I do not think this point justifies another round of revision. In my view, the paper is ready for publication. However, I believe that a short statement acknowledging the typical range of uncertainty in fracture energy estimates would improve the transparency and reliability of the work, especially for non-specialist readers.

A: We thank the reviewer for clarifying their viewpoint. As suggested, we have revised the main text (lines 242–246 and 258–262) to clarify how our fracture energy estimates compare with, and fall within the uncertainties of, previous seismological, modeling, and experimental studies.

R3 – Minor suggestions

R3: *Figure 7: The reported peak friction values of 0.30–0.36 are difficult to visually confirm. I suggest increasing the line width or using a more distinct color/contrast to make the friction peak more visible.*

A: We thank the reviewer for this observation. However, we note that the same issue was raised by Reviewer 2 in the previous round of revision. While increasing line width would improve the visibility of the peak friction detail, it would also render the entire graph overly bold and visually bulky. To address this concern, we previously included Supplementary Figure 9, which presents the experimental dataset in greater detail and allows specialist readers to examine the evolution of friction around its peak values.

R3 – LEFM model explanation: *In Section B of the Methods, I recommend adding a short sentence at the beginning to explain that, for dynamic rupture to occur, the energy flux $G\dot{G}$ must equal the energy required to fracture the surface Γ Γ . This would help readers who are less familiar with fracture mechanics understand the physical basis of the model.*

A: Following the reviewer's suggestion, we have revised Section B of the Methods (lines 488–490 and 498–500) to improve clarity.

R3 – Supplementary Table 1: *Please consider adding the calculated energy values for the principal slip zone (PSZ) to Supplementary Table 1. Currently, these data are not directly accessible to the reader and would be useful for completeness and reproducibility.*

A: We have revised Table 1 and added the calculated values of fracture energy of the principal slip zone from the experimental data of each experiment using Equation 2 in Supplementary Method A – Estimation of Fracture Energy.

R3 – Overall, *I appreciate the authors' careful work on this revised version. I am satisfied with the manuscript as it now stands and believe that these final clarifications will further strengthen its impact.*

A: We thank R3 for their appreciation of our effort during the final revision of our manuscript. By addressing their final comments we believe that we manage to improve the clarity of certain aspects of our manuscript.

Reviewer #3 (Remarks on code availability):

R3 – *I used the code for reproducing Fig. 10. I added an additional comment to the authors to add the energy calculations in Table 1 because the raw experimental data is not available.*

A: This has now been addressed as requested by R3, see previous comment.